# Disentangled generative models for robust dynamical system prediction

## Abstract

Deep neural networks have become increasingly of interest in dynamical system prediction, but out-of-distribution generalization and long-term stability still remains challenging. In this work, we treat the domain parameters of dynamical systems as factors of variation of the data generating process. By leveraging ideas from supervised disentanglement and causal factorization, we aim to separate the domain parameters from the dynamics in the latent space of generative models. In our experiments we model dynamics both in phase space and in video sequences and conduct rigorous OOD evaluations [1]. Results indicate that disentangled VAEs adapt better to domain parameters spaces that were not present in the training data. At the same time, disentanglement can improve the long-term and out-of-distribution predictions of state-of-the-art models in video sequences.[2]

## 1 Introduction

The robust prediction of dynamical systems behaviour remains an open question in machine learning, and engineering in general. The ability to make robust predictions is important not only for forecasting systems of interest like weather (Garg et al., 2021; Ravuri et al., 2021) but even more so because it enables innovations in fields like system control, autonomous planning (Hafner et al., 2018) and computer aided engineering (Brunton et al., 2020). In this context, the use of deep generative models has recently gained significant traction for sequence modelling (Girin et al., 2020).

Robustness of machine learning models can be considered along two axes: long-term prediction and out-of-distribution (OOD) performance. Accurate long-term prediction can be notoriously difficult in many dynamical systems, because error accumulation can diverge in finite time (Zhou et al., 2020; Raissi et al., 2019), a problem that even traditional solvers can suffer from. More importantly, machine learning techniques are known to suffer from poor OOD performance (Goyal & Bengio, 2020), i.e. when they are employed in a setting they had not encountered during the training phase.

Before addressing the OOD problem, we must first define what constitutes as OOD in dynamical systems. We start by the observation that even simple dynamical systems, i.e the swinging pendulum or the $n$-body system, can have multiple continuous parameters that affect their evolution. These parameters can be manifested as differential equation coefficients, boundary or initial conditions etc. Our starting point is to consider distinct ranges of those parameters as separate domains. Under this view, it becomes apparent why OOD prediction of dynamical systems can be hard: capturing the whole range of those parameters in a single training set is unrealistic (Fotiadis et al., 2020) and further inductive biases are required (Miladinović et al., 2019; Bird & Williams, 2019; Barber et al., 2021). From a dynamical system point of view, different parameters can produce widely different trajectories in phase space. A motivating example can be bifurcations which occur when a small change in the parameters of a system causes a sudden qualitative change in its behaviour.

We focus on the inductive bias of disentangled representations for which the dynamics are separated from the domain parameters. Many approaches based on the use of neural networks try to jointly learn the dynamics and the physical parameters, which results in convoluted representations and usually leads to overfitting (Bengio et al., 2012). System identification can be used to extract parameters, but

---

[1]Code for reproducing our experiments at: `https://anonymous.4open.science/r/dis-dyn-systems/`

[2]Animated phase-space and video predictions are available at: `https://bit.ly/dis-dyn-systems`

requires knowledge of the underlying system to be computationally effective (Ayyad et al., 2020). We, instead, leverage advances in Variational Autoencoders (VAEs) (Kingma & Welling, 2014) that enable learning disentangled representations. Disentanglement enables different latent variables to focus on different factors of variation of the data distribution, and has been applied in the context of image generation (Higgins et al., 2017; Kim & Mnih, 2018). This can be extended to modelling dynamical systems by looking at disentanglement from a causal perspective: from all the generative models which can have the same marginal distribution, identify the one with the true causal factors. To map this idea to sequence modelling we treat the domain parameters of a dynamical system as factors of variation. Recent findings (Locatello et al., 2018) emphasize the vital role of inductive biases from models or data for useful disentanglement. Unsupervised disentanglement, based on the assumption of domain stationarity, is a promising direction (Miladinović et al., 2019; Li & Mandt, 2018). Nevertheless, this leaves a wealth of ground truth domain parameters, which can be cheaply collected in simulated datasets. This type of privileged information originating from simulations has been shown to be effective for domain adaptation in computer vision tasks (Sarafianos et al., 2017; Lee et al., 2018). We thus use supervised disentanglement (Locatello et al., 2019) by leveraging the ground truth domain parameters. To the best of our knowledge, using domain parameters information this way, has not been previously explored in the dynamical system prediction setting.

**Contributions** While others have treated domain parameters as factors of variation in the data distribution, our work is the first, to the best of our knowledge, that explicitly uses privileged information from simulated data to disentangle those domain parameters from dynamics in a supervised way. We furthermore conduct experiments both in the low-dimensional phase space of 3 dynamical systems and the high-dimensional video rendering of a swinging pendulum. Disentanglement has, in the past, been mostly applied to VAEs because they are easily amenable to it. We additionally apply disentanglement on a more powerful, hybrid, model with both stochastic and deterministic parts (Hafner et al., 2018). In doing so, we not only assess disentanglement on a generative model outside boundaries of VAEs but furthermore we do it on a model which is considered state-of-the-art in long-term video prediction (Saxena et al., 2021). In all cases, the prediction performance is assessed both in-distribution and also in OOD settings of increasing degrees of distribution shift. To our understanding, this is the first time such a rigorous OOD test is performed. Our results in phase-space demonstrate that disentangled models can better capture the variability of dynamical systems compared to baseline models both in-distribution and OOD. In modelling dynamics in video sequences, results indicate that disentanglement is beneficial both for long-term prediction and OOD prediction.

**Limitations** This work focuses on dynamical system prediction. While the results can potentially open up many applications in general time-series modelling, this is out of the scope of this work. We prioritize to empirically study OOD downstream task performance and the inspection of the disentangled representations with appropriate metrics is left out of scope in this work.

## 2 Related Work

**VAEs and disentanglement** Disentanglement aims to produce representations where separate factors of variation in the data are encoded into independent latent components. This can be seen as finding the true causal model of the data. While supervised disentanglement in generative models is a long-standing idea (Mathieu et al., 2016), information-theoretic properties can be leveraged to allow unsupervised disentanglement in VAEs (Higgins et al., 2017; Kim & Mnih, 2018). The impossibility result from (Locatello et al., 2018) suggested that disentangled learning is only possible by inductive biases coming either from the model or the data. Hence, the focus shifted back to semi- or weakly-supervised disentanglement approaches (Locatello et al., 2019; 2020). While most of these methods focus on disentanglement metrics, we opt to directly assess using a downstream prediction task.

**Disentanglement in sequence modelling** While disentanglement techniques are mainly tested in a static setting, there is a growing interest in applying it to sequence dynamics. Using a bottleneck based on physical knowledge, Iten et al. (2018) learn an interpretable representation that requires conditioning the decoder on time, but it can return physically inconsistent predictions in OOD data (Barber et al., 2021). Deep state-space models (SSMs) have also employed techniques for disentangling content from dynamics (Fraccaro et al., 2017; Li & Mandt, 2018), but, focus mostly on modelling variations in the content, failing to take dynamics into account. In hierarchical approaches (Karl et al., 2017), different layers of latent variables correspond to different timescales: for example,

Figure 1: **The VAE-SD model (Left).** From an $n$-dimensional phase space input, a $o$-dimensional prediction of future time-steps is derived. The loss function has three parts: the reconstruction loss is replaced by a prediction loss, the KL-divergence enforces the prior on to the latent space, and the extra loss term enforces the supervised disentanglement of domain parameters in latent space. **The CNN-VAE-SD model(Right).** The input frames are first encoded to a low-dimensional space, analogous to phase space. Then, as before the VAE-SD prediction scheme is applied recursively. The low-dimensional predictions are then decoded back to pixel space.

in speech analysis for separating voice characteristics and phoneme-level attributes (Hsu et al., 2017). In an approach similar to our work, Miladinović et al. (2019) separate the dynamics from sequence-wide properties in dynamical systems like Lotka-Volterra, but do so in an unsupervised way which dismisses a wealth of cheap information and only assesses OOD generalization in a limited way.

**Feed-forward models for sequence modelling** Deep SSM models are difficult to train as they require non-trivial inference schemes and a careful design of the dynamic model (Krishnan et al., 2015; Karl et al., 2017). Feed-forward models, with necessary inductive biases, have been used for sequence modelling in dynamical systems (Greydanus et al., 2019; Fotiadis et al., 2020). In works like Hamiltonian Neural Networks Greydanus et al. (2019) the domain is fixed; together with Barber et al. (2021), our work is an attempt in tackling domain variability.

**Privileged information for domain adaptation.** Using privileged information during training has been shown to help with domain adaptation in computer vision tasks. Using segmentation masks of simulated urban scenery can improve semantic segmentation on the target domain (Lee et al., 2018), while clip art data can help with domain transfer in an action recognition task (Sarafianos et al., 2017).

## 3 METHODS

### 3.1 VARIATIONAL AUTOENCODERS

Variational autoencoders (VAEs) (Kingma & Welling, 2014) offer a principled approach to latent variable modeling by combining a variational inference model $q_\phi(z|x)$ with a generative model $p_\theta(x|z)$. As in other approximate inference methods, the goal is to maximize the evidence lower bound (ELBO) over the data:

$$\mathcal{L}_{\phi,\theta}(\boldsymbol{x}) = \mathbb{E}_{q_\phi(\mathbf{z}|\mathbf{x})}[\log p_\theta(\mathbf{x} \mid \mathbf{z})] - D_{KL}(q_\phi(\mathbf{z} \mid \mathbf{x})||p(\mathbf{z})) \qquad (1)$$

The first part of the ELBO is the reconstruction loss (in our case the prediction loss) and the second part is the Kullback-Leibler divergence that quantifies how close the posterior is to the prior.

**Design choices for the model** We use an isotropic unit Gaussian prior $p(z) = \mathcal{N}(z \mid \mathbf{0}, \boldsymbol{I})$ which helps to disentangle the learned representation (Higgins et al., 2017). The approximate posterior (encoder) distribution is a Gaussian with diagonal covariance $q_\phi(z \mid x) = \mathcal{N}(z \mid \boldsymbol{\mu}_z, \boldsymbol{\Sigma}_z)$, allowing a closed form KL-divergence, while the decoder has a Laplace distribution $p_\theta(x \mid z) = $ Laplace $(x \mid \boldsymbol{\mu}_x, \gamma \boldsymbol{I})$ with constant diagonal covariance $\gamma > 0$, which is tuned empirically. This leads to an $L_1$ loss that provides improved results in some problems (Mathieu et al., 2018) and empirically works better in our case. The parameters $\boldsymbol{\mu}_z \equiv \boldsymbol{\mu}_z(x; \phi), \boldsymbol{\Sigma}_z \equiv \text{diag} \left[ \boldsymbol{\sigma}_z(x; \phi) \right]^2$, and $\boldsymbol{\mu}_x \equiv \boldsymbol{\mu}_x(z; \theta)$ are computed via feed-forward neural networks.

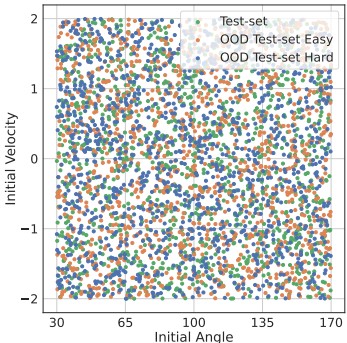 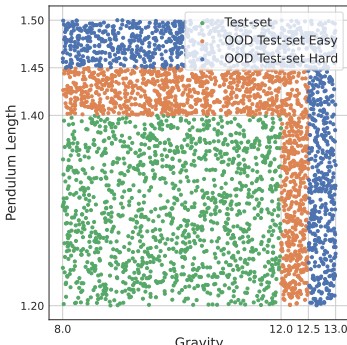

Figure 2: **Parameter distribution for the video pendulum test-sets.** The initial angle and angular velocity are drawn from the same uniform distribution for all test-sets. For the in-distribution test-set we draw the pendulum length and gravity from the same distribution as during training. The OOD test-sets represent distribution shifts of increasing magnitude, where parameters are drawn from totally different space which has zero overlap with the training and in-distribution test-set.

## 3.2 DISENTANGLEMENT OF DOMAIN PARAMETERS IN LATENT SPACE

Apart from the disentanglement that stems from the choice of prior $p(z)$, we explicitly disentangle part of latent space so that it corresponds to the domain parameters of each input sequence. We achieve this by using a regression loss term $\mathcal{L}_\xi(\mathbf{z}_{1:k}, \xi)$ between the ground truth factors of the domain parameters $\xi \in \mathbb{R}^k$ and the output of the corresponding latents, $\mathbf{z}_{1:k}$. We, empirically, opted for an $L_1$ loss, corresponding to a Laplacian prior with mean $\xi$ and unitary covariance. Previous methods have reported that binary cross-entropy works better than $L_2$ (Locatello et al., 2019) but this does not fit well in a setting like ours. We hypothesize that BCE works better because of the implicit scaling. To address this, we propose applying a function $\mathcal{G}(\mu_{z_i})$ which linearly scales the $\mu_{z_i}$ between the min and max values of the corresponding factor of variation:

$$\mathcal{G}\left(\mu_{z_i}\right) = \mu_{z_i} \cdot (\max(\xi_i) - \min(\xi_i)) + \min(\xi_i) \tag{2}$$

where $\xi_i$ are the domain parameters and their corresponding minimum and maximum values of domain parameters from the training set. In all cases, the regression term is weighted by a parameter $\delta$ which is empirically tuned. Plugging these choices in results in the following loss function:

$$
\begin{aligned}
\mathcal{L}_{\phi,\theta}(\boldsymbol{x}) = \mathbb{E}_{q_\phi(\boldsymbol{z}|\boldsymbol{x}_{1:n})} & \left[ \frac{1}{\gamma} \|\boldsymbol{x}_{n+1:n+o} - \boldsymbol{\mu}_x(\boldsymbol{z};\theta)\|_1 \right] + d \log \gamma && \text{(Prediction loss)} \\
& + \|\boldsymbol{\sigma}_z(\boldsymbol{x}_{1:n};\phi)\|_2^2 - \log\left|\text{diag}\left[\boldsymbol{\sigma}_z(\boldsymbol{x}_{1:n};\phi)\right]^2\right| + \|\boldsymbol{\mu}_z(\boldsymbol{x}_{1:n};\phi)\|_2^2 && \text{(KL-Divergence)} \\
& + \delta \left\|\boldsymbol{\xi}_x - \mathcal{G}\left(\boldsymbol{\mu}_{z_{1:k}}(\boldsymbol{x}_{1:n};\phi)\right)\right\|_1 \Big\} && \text{(Sup. disentangl. loss)}
\end{aligned}
\tag{3}
$$

Using the reparameterization trick (Kingma & Welling, 2014), the loss is amenable to optimization by stochastic gradient descent, with batch size $n$. The model architecture can be seen in Figure 1(left).

## 3.3 DISENTANGLEMENT FOR VIDEO DYNAMICS

We further investigate the effect of disentanglement in video sequence dynamics. To this end, two generative models are used. The first is derived from the VAE formulation of the previous section and is called CNN-VAE and is similar to the VAE with the addition of a convolutional encoder and a decoder. The encoder projects the input frames down to a low-dimensional space which can be thought as equivalent to the phase space of the system. A VAE is applied in this projection to predict in the future coordinates in the "phase space". The decoder then maps the predictions of the VAE back to pixel space. The schematic of the model can be seen in Figure 1(right).

The second model we use is the Recurrent State Space Model (RSSM) which has been successfully used for planning (Hafner et al., 2018). Since RSSM is a hybrid model combining deterministic and variational components, it allows us to assess disentanglement outside the limited scope of VAEs. Furthermore, using a state-of-the-art model in long-term video prediction (Saxena et al., 2021), allows

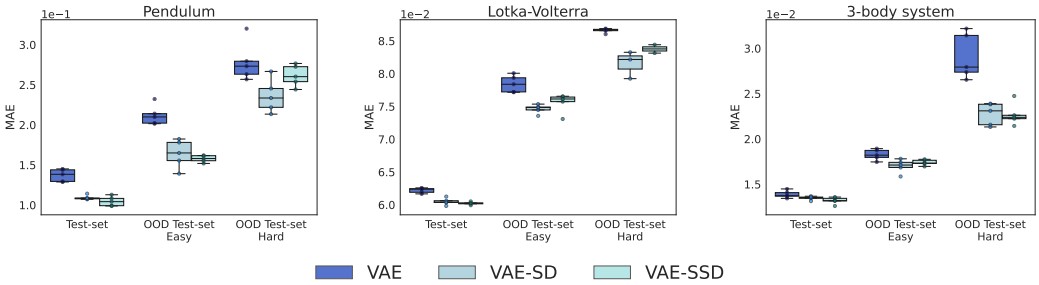

Figure 3: **Disentanglement scaling methods.** MAE for the first 200 time-steps. Boxplots show the top 5 best performing sets of hyperparameters of each architecture. Both VAE-SD & VAE-SSD outpeform the VAE in all 3 systems. VAE-SSD better captures the parameter space of the original test-set but in most cases VAE-SD generalizes better OOD.

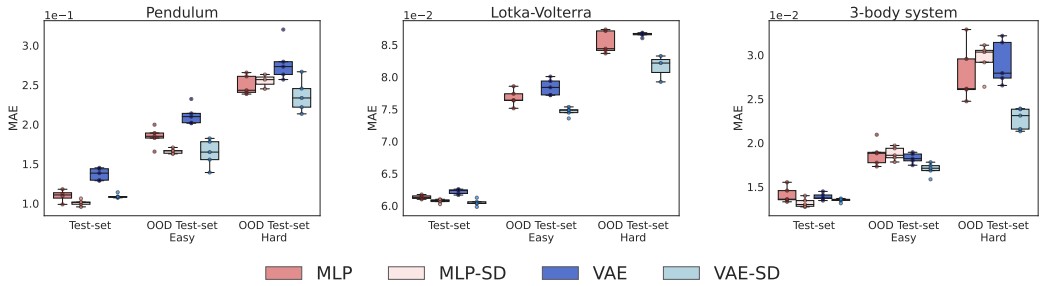

Figure 4: **Disentanglement in VAE vs MLP.** MAE for the first 200 time-steps. Boxplots show the top 5 best performing sets of hyperparameters of each architecture. While disentanglement in VAE-SD consistently improves results, disentanglement in MLP-SD does not always generalize well OOD, producing unstable predictions for the OOD Lotka-Volterra datasets.

us to identify the limits of applying disentanglement in competitive models. The loss function we use shares the same formulation as in the original work of Hafner et al. (2018) with the addition of the supervised disentanglement loss. Since in the RSSM formulation there are latent variables for each time-step, we apply a disentanglement loss on all of them, which empirically is set to be $L_2$:

$$\mathcal{L}_{RSSM-SD} = \sum_{t=1}^{T} \left( \underbrace{\mathbb{E}_{q(s_t|o_{\leq t})}[\ln p(o_t \mid s_t)]}_{\text{reconstruction}} - \underbrace{\mathbb{E}_{q(s_{t-1}|o_{\leq t-1})}\left[\text{KL}[q(s_t \mid o_{\leq t})\|p(s_t \mid s_{t-1})]\right]}_{\text{prediction}} \right.$$
$$\left. + \underbrace{\delta\mathbb{E}_{q(s_t|o_{\leq t})}\left[\left\|\boldsymbol{\xi} - s_t^{(1:k)}\right\|_2\right]}_{\text{supervised disentanglement loss}} \right) \tag{4}$$

Where $o_t$ is the observations, $s_t$ the stochastic latent variables at time $t$, $\boldsymbol{\xi}$ are the $k$ dimensional domain parameters and $\delta$ tunes the supervised disentanglement strength.

## 4 EXPERIMENT - ODE PHASE SPACE DYNAMICS

### 4.1 DATASETS

In the phase-space experiments we compare the models on three well studied dynamical systems, the swinging pendulum, the Lotka-Volterra equations used to model prey-predator populations, and the planar 3-body system:

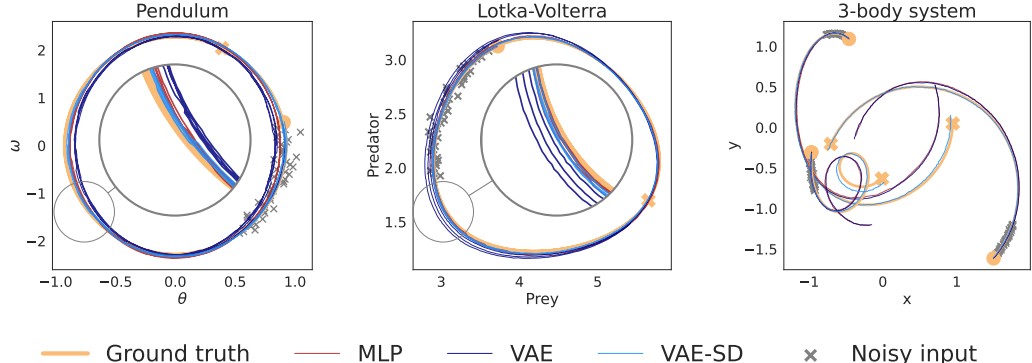

Figure 5: **Model predictions in phase space**. Trajectories are taken from the OOD Test-Set Hard of each system. The model input is noisy. The circle and bold '×' markers denote the start and end of the ground truth trajectories respectively.

**Simple pendulum:** $\ddot{\theta} + \frac{g}{\ell}\sin\theta = 0$

**Lotka-Volterra:** $\begin{aligned}\dot{x} &= \alpha x - \beta xy \\ \dot{y} &= \delta xy - \gamma y\end{aligned}$

**3-body system:** $\begin{aligned}\bar{m}_i \frac{d\vec{v}_i}{dt} &= K_1 \sum_j \frac{\bar{m}_i \bar{m}_j}{\bar{r}_{ij}^3}\overrightarrow{r_{ij}} \\ \frac{d\vec{\bar{x}}_i}{d\bar{t}} &= K_2\vec{v}_i, i \in 1, 2, 3\end{aligned}$

The systems where chosen for varied complexity in terms of degrees of freedom, number of ODE equations and factors of variation. For the pendulum we consider one factor of variation, its length $l$; Lotka-Volterra has 4 factors of variation $\alpha, \beta, \gamma, \delta$ and the 3-body system has also 4 factors of variation $K_1, m_1, m_2, m_3$. Factors are drawn uniformly from a predetermined range which is the same between the training, validation and test sets. To further assess the OOD prediction accuracy, we create two additional test sets with factor values outside of the original range. We denote these datasets as OOD Test-set Easy and Hard, representing a smaller and bigger shift from the original range. As a visual example, the distribution of the factors of variation for the Lotka-Volterra system is illustrated in Figure 9 of the Appendix. The data were additionally corrupted with Gaussian noise. Dataset details can be found on Table 1 of the Appendix.

## 4.2 MODELS AND TRAINING

The main goal of these experiments is to assess whether OOD prediction can be improved by disentangling dynamical system parameters in the latent space of VAEs. We opt to use simple models to allow more experiments and comparisons. Our main baseline is the VAE upon which we propose two enhancements that leverage supervised disentanglement. The first VAE model, termed VAE-SD uses supervised disentanglement without a scaling function while the second model termed VAE-SSD uses an additional linear scaling function $\mathcal{G}(\mu_{z_i})$ for the latent variable mean vector $\mu_{z_i}$, as described in Section 3.2. Another baseline is a multilayer perceptron (MLP) autoencoder which allows comparison with a deterministic counterpart of the VAE. We also use supervised disentanglement on the latent neurons of the MLP, a model we refer to as MLP-SD. This enables us to assess if the privileged information can improve deterministic models. Lastly, we include an LSTM model, a popular choice for low dimensional sequence modelling (Yu et al., 2019), as a representative recurrent method.

Early experiments revealed a significant variance on the performance of the models, depending on hyperparameters. Under these conditions, we took various steps to make model comparisons as fair as possible. Firstly, all models have similar capacity in terms of neuron count. Secondly, we tune various hyperparameter dimensions, some of which are shared, while others are model-specific. Third, we conduct a thorough grid search on the hyperparameters to avoid undermining a model (details can be found in Tables 3, 4 and 5 of the Appendix). Lastly, we train the same number of experiments for all models which amounts to 1440 trained models in total, as summarized in Table 2 of the Appendix.

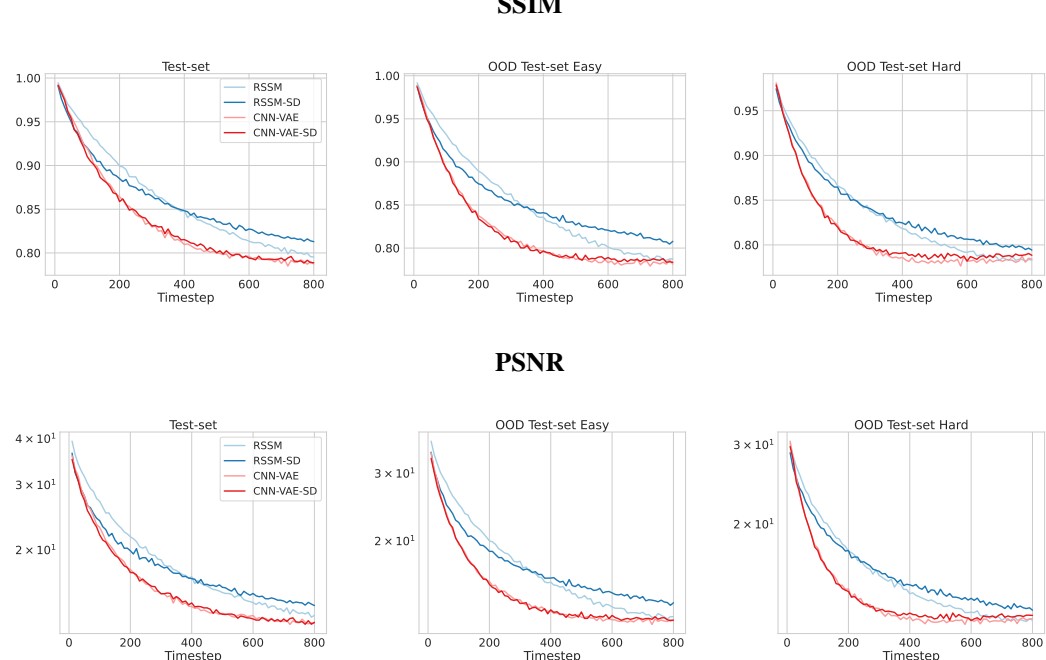

Figure 7: **Prediction quality on the video pendulum.** SSIM(top) and PSNR(bottom) as a function of the distance predicted into the future (x axis)

### 4.3 RESULTS

For each dynamical system we focus on the performance on the three test-sets, the in-distribution test set, which shares the same parameter distribution with the training set, and the two OOD test-sets (Easy and Hard), which represent an increasing parameter shift from the training data. Models are compared on the cumulative Mean Absolute Error (MAE) between prediction and ground truth for the first 200 time-steps. We consider this to be sufficiently long-term as it is at least 20 times longer than the prediction horizon used during training. Long predictions are obtained by re-feeding the model predictions back as input. This approach has been shown to work well in systems where the dynamics are locally deterministic (Fotiadis et al., 2020). A summary of the quantitative results can be found in Figures 3 & 4 and Table 8. To account for the variability in the results, we present a summary of the best 5 runs of each model, selected by validation MAE. We generally observe that model performance is correlated to the distribution shift of test-sets, and this is consistent for all systems and models. The MAE is increasing as we move from the in-distribution test-set to the OOD Easy and Hard test-sets. Nevertheless, not all models suffer equally from the OOD performance drop.

Comparing the VAEs (Figure 3), we see that disentangled VAE models offer a substantial and consistent improvement over the VAE across all 3 dynamical systems. The improvement is more pronounced for the OOD test-sets where the distribution shift is greater, a strong indication that disentanglement of domain parameters is an inductive bias that can lead to better generalization. We also observe that VAE-SSD models the in-distribution data better that VAE-SD. This seems to come at a slight overfitting cost, because the VAE-SD provides better OOD extrapolation in most cases. This could be explained because the scaling function is dependent on min and max values of the factors of the training set. The extra information allows the model to better capture the training data but sacrifices some generalization capacity.

On the other hand, disentanglement results for the MLP are mixed. While in-distribution MLP-SD offers better results than the plain MLP, on the OOD test-sets, MLP-SD only performs favourably in the pendulum data. Furthermore in Lotka-Volterra, MLP-SD models are very unstable, and this is a drawback that affects some VAE-SD model too (see Table 9 of the Appendix). Probabilistic models seem better suited to capture the variation in the data. The contrast between VAE-SD and MLP-SD illustrates that making use of privileged information and latent space disentanglement are not trivial

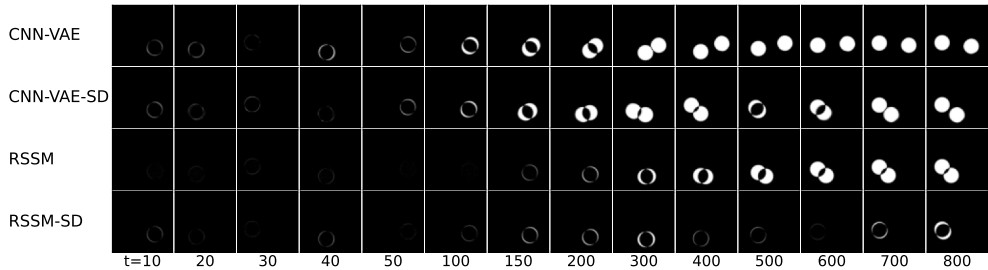

Figure 8: Absolute difference between ground truth and predictions on the test-set of the pendulum data set.

and more work is needed to help us understand what works in practice and why. Lastly, the LSTM (Figure 11 & Table 8 of the Appendix) is only comparable in the pendulum dataset and only for small OOD shifts. Qualitative predictions can be found in Figure 5.

## 5 EXPERIMENT - VIDEO SEQUENCE DYNAMICS

In the first experiment we assessed supervised disentanglement for phase space prediction, where the states of the input trajectories are fully observable and only the domain parameters are unknown. This experiment extends the idea of supervised disentanglement to pixel-space input and output, where the physical states have to be inferred by the model.

### 5.1 DATASETS

The dynamical system we use in this experiment is the swinging pendulum, a common benchmark for modelling dynamics in video sequences (Brunton et al., 2020; Barber et al., 2021). We consider 4 factors of variation, the length $l$, gravity $g$ and initial angle $\theta$ and angular velocity $\omega$. Factors are drawn uniformly from a predetermined range. As before, we create a test-set and two additional OOD test-sets (Easy and Hard). The OOD sets have length and gravity values outside of the original range, while the initial conditions $\theta, \omega$ are drawn from the same distribution. The distribution of the factors of variation for the test-sets is illustrated in Figure 2. The trajectories are first computed in phase space using a numerical simulator and then rendered as video frames of $64 \times 64$ pixels. More details about the dataset can be found in Section A.2 of the Appendix.

### 5.2 MODELS AND TRAINING

In this experiment we use two different models CNN-VAE and RSSM. CNN-VAE is described in Section 3.3 and architectural details can be found in Section B.2.1. During training the CNN-VAE the inner VAE is recursively used to predict, the number of recursions being a hyperparameter (Table 6 of the Appendix). We found that this type of training leads to more stable long term predictions. In total, 48 CNN-VAE models are trained half of which are with supervised disentanglement (CNN-VAE-SD). The RSSM model is a generative model including both a stochastic and deterministic component. We only use supervised disentanglement on the stochastic part, and term that model RSSM-SD. Disentanglement is applied all four factors of variation of the domain, despite only length and gravity varying between datasets. Detailed architectural and training details can be found in Section B.2.2 of the Appendix.

### 5.3 RESULTS

Figure 7 shows the quality of predictions on video pendulum on two metrics: structural similarity (SSIM) and peak signal-to-noise ratio (PSNR) as a function of predicted time distance. We select the models which have the best cumulative metrics over the first 800 timesteps on a validation set.

For the CNN-VAE, effects of disentanglement are more subtle. We observe that, in-distribution, the disentangled CNN-VAE-SD has very similar quality when compared to the CNN-VAE. For the OOD

datasets, though, disentanglement offers improved long-term predictions. The improvement is more noticeable on the OOD test-sets, indicating that disentanglement can help with OOD robustness. For RSSM, we first note that both models perform significantly better than the CNN-VAE models, which is expected since they are considered competitive in long-term video prediction. Disentanglement in RSSM seems to produce a trade-off. The plain RSSM model better in short-term prediction but its performance deteriorates with time, reaching VAE-CNN levels in all metrics. On the other hand, the RSSM-SD model provides the best long-term scores in all metrics and all datasets. Qualitative results in Figure 5 show that almost all models produce accurate short time predictions (approximately up to 200 time-steps). This further stresses the importance of disentanglement for long-term performance. In terms of OOD robustness, disentanglement also appears to be helping. While the RSSM-SD model lacks in short-term prediction quality on the in-distribution test-set, this performance gap closes as the OOD test-sets get harder. More specifically, on the in-distribution test-set the RSSM-SD overtakes RSSM in SSIM after around 400 frames, while in the OOD Easy and Hard test sets, this happens around 350 and 250 time-steps respectively. This narrowing gap indicates robustness improves with increasing distribution shifts. The above findings are corroborated by LPIPS (Zhang et al., 2018) comparisons (Figure 13 and Table 10 of the Appendix). Furthermore, the qualitative results show that all models accurately capture the appearance of the pendulum even long-term. Where they differ is on how well they capture the dynamics of the pendulum movement. This could offer an explanation why disentangling the domain from the dynamics is important, and why in practice offers better long-term and out-of-distribution performance.

Overall, experiments suggest that supervised disentanglement can be used to model dynamical systems in video sequences, resulting in improved long-term and OOD performance.

## 6 CONCLUSIONS

Using supervised disentanglement of domain parameters in generative models is a promising avenue for improving robustness. Our experiments show that it can improve both OOD generalization and long-term prediction of dynamical systems. This was demonstrated in phase-space with VAEs and also in video sequence modelling with state-of-the-art RSSMs.

By treating the domain parameters as factors of variation of the data and applying supervised disentanglement, several inductive biases are potentially enforced. First, the model in addition to prediction also performs "soft" system identification which acts as a regularizer. Second, it creates an implicit hierarchy such that some latent variables correspond to sequence-wide domain parameters and the rest capture instant dynamics. We speculate that this could additionally make the latent space more interpretable. Third, if the model can correctly extract the parameters this mean that the prediction is based on both of them which is closer to how numerical integrators work, where the domain is known. All of these could lead the model to learn the correct causal structure of the data. Nevertheless, using privileged information for OOD robustness is not always straightforward and requires further exploration. This is evident by the results of the MLP autoencoders which do not yield as consistent improvements. A criticism of our method could be that cheap privileged information is not always available and/or depends on using simulated data. Firstly, training on simulations is an increasingly appealing option because it is a cheap way to generate data to begin with. This is, also, clearly demonstrated by the many advancements on techniques like sim2real (Peng et al., 2017) that try to bring models trained in simulated data to the real world. So there seems to be no reason not to use the privileged information that comes with simulated data. Under that light supervised disentanglement can provide a pathway for real world applications where robustness in dynamical system prediction is critical. Applying the method to other datasets where there are more complex dynamic can increase its relevance. Sequence-wide parameters could also be exploited through self-supervision.

### REPRODUCIBILITY STATEMENT

We provide all the necessary code to reproduce our experiments at the anonymous repo `https://anonymous.4open.science/r/dis-dyn-systems` (will be de-anonymized after the review process). The repo contains code for generating all the datasets from scratch and also code for training all the models presented in this work. The README also contains instructions on how to train the models. The hyperparameters we have used are clearly and thoroughly presented in the

Appendix. These steps should significantly help others reproduce our experiments. For any further clarifications, you are encouraged to contact the corresponding author(s).

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

## APPENDIX

## A   DATASETS

### A.1   PHASE SPACE

For simulations, we use an adaptive Runge-Kutta integrator with a timestep of 0.01 seconds. Each simulated sequence has a different combination of factors of variation. Simulation of the pendulum uses an initial angle $\theta$ which is randomly between $10^o - 170^o$ while the angular velocity $\omega$ is 0. For the other two systems the initial conditions are always the same to avoid pathological configurations.

Table 1: **Datasets.** In L-V and 3-body OOD test sets, at least one domain parameter is outside of the parameter range used for training.

| | Pendulum | Lotka-Volterra | 3-Body |
|---|---|---|---|
| ODEs | $\ddot{\theta} + \frac{g}{\ell}\sin\theta = 0$ | $\dot{x} = \alpha x - \beta xy$ 
 $\dot{y} = \delta xy - \gamma y$ | $\bar{m}_i \frac{d\vec{v}_i}{dt} = K_1 \sum_j \frac{\bar{m}_i \bar{m}_j}{\bar{r}_{ij}^3} \overrightarrow{r_{ij}}$ 
 $\frac{d\vec{x}_i}{dt} = K_2 \vec{v}_i$ |
| Number of ODEs | 1 | 2 | 6 |
| Independent Variables | $\theta, \omega$ | $x$(prey), $y$(predator) | $\overrightarrow{x}_i, \overrightarrow{v}_i, i = 1, 2, 3$ |
| Initial values | $\theta \in [10^o - 170^o]$ 
 $\omega = 0$ | $x = 5, y = 3$ | $\overrightarrow{x_1} = (-1, -1), \overrightarrow{v_1} = (0.0, 0.5)$ 
 $\overrightarrow{x_2} = (1, -1), \overrightarrow{v_2} = (0.5 - 0.5)$ 
 $\overrightarrow{x_3} = (0, 1), \overrightarrow{v_3} = (-0.5, 0.0)$ |
| Timestep | 0.01 | 0.01 | 0.01 |
| Sequence length | 2000 | 1000 | 1000 |
| Noise $\sigma^2$ | 0.05 | 0.05 | 0.01 |
| Factors of variation | $l$(length) | $\alpha, \beta, \gamma, \delta$ | $K_2, m_1, m_2, m_3$ |
| Train/Val/Test | $l \in [1.0 - 1.5]$ | $A = \{\alpha \in [1.95, 2.05]\}$ 
 $B = \{\beta \in [0.95, 1.05]\}$ 
 $C = \{\gamma \in [3.95, 4.05]\}$ 
 $D = \{\delta \in [1.95, 2.04]\}$ 
 $\Omega_{\text{train}} = (A \times B \times C \times D)$ | $K = \{K_2 \in [1.95, 2.05]\}$ 
 $M1 = \{m_1 \in [1.95, 2.05]\}$ 
 $M2 = \{m_2 \in [1.95, 2.05]\}$ 
 $M3 = \{m_3 \in [1.95, 2.05]\}$ 
 $\Omega_{\text{OOD Hard}} =$ 
 $(K \times M1 \times M2 \times M3)$ |
| OOD Test Set Easy | $l \in [1.5 - 1.6]$ | $A = \{\alpha \in [1.94, 2.06]\}$ 
 $B = \{\beta \in [0.94, 1.06]\}$ 
 $C = \{\gamma \in [3.94, 4.06]\}$ 
 $D = \{\delta \in [1.94, 2.06]\}$ 
 $\Omega_{\text{OOD Easy}} = (A \times B \times C \times D) \setminus \Omega_{\text{train}}$ | $K = \{K_2 \in [1.94, 2.06]\}$ 
 $M1 = \{m_1 \in [1.94, 2.06]\}$ 
 $M2 = \{m_2 \in [1.94, 2.06]\}$ 
 $M3 = \{m_3 \in [1.94, 2.06]\}$ 
 $\Omega_{\text{OOD Hard}} =$ 
 $(K \times M1 \times M2 \times M3) \setminus \Omega_{\text{train}}$ |
| OOD Test Set Hard | $l \in [0.9 - 1.0]$ | $A = \{\alpha \in [1.93, 2.07]\}$ 
 $B = \{\beta \in [0.93, 1.07]\}$ 
 $C = \{\gamma \in [3.93, 4.07]\}$ 
 $D = \{\delta \in [1.93, 2.07]\}$ 
 $\Omega_{\text{OOD Hard}} =$ 
 $(A \times B \times C \times D) \setminus (\Omega_{\text{train}} \cup \Omega_{\text{OOD Easy}})$ | $K = \{K_2 \in [1.93, 2.07]\}$ 
 $M1 = \{m_1 \in [1.93, 2.07]\}$ 
 $M2 = \{m_2 \in [1.93, 2.07]\}$ 
 $M3 = \{m_3 \in [1.93, 2.07]\}$ 
 $\Omega_{\text{OOD Hard}} =$ 
 $(K \times M1 \times M2 \times M3) \setminus (\Omega_{\text{train}} \cup \Omega_{\text{OOD Easy}})$ |
| Number of sequences 
 Train/Val/Test | | 8000/1000/1000 | |
| OOD Test Set Easy | | 1000 | |
| OOD Test Set Hard | | 1000 | |

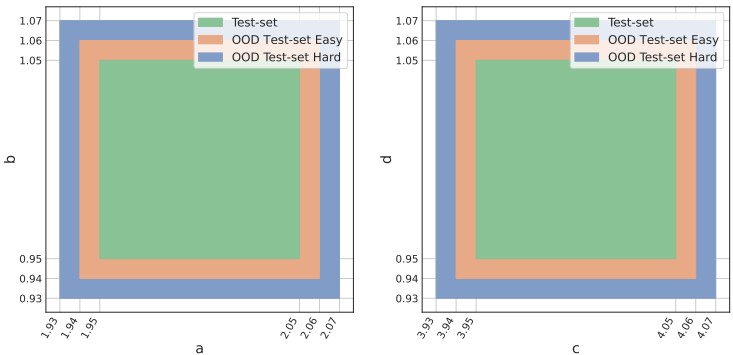

Figure 9: Example illustration of the parameter distribution for the LV test sets. The regions do not overlap, colors represent regions not boundaries. The OOD Easy test (orange) set does not include any of the parameter configurations of the training and original test set (green). Respectively, the OOD Hard dataset (blue) does not include none of the OOD Easy or the original test set configurations. The parameter space of the blue region is almost half as big at the green area (again without any overlap), signifying a significant OOD shift).

## A.2 VIDEO PENDULUM

This data set contains image sequences of a moving pendulum under different conditions. The positions of the pendulum are first computed by a numerical simulator and then rendered in pixel space as frames of dimension $64 \times 64$. An example image sequence is shown in Figure 10. For the simulations, we use an adaptive Runge-Kutta integrator with a timestep of 0.05 seconds. The length of the pendulum, the strength of gravity and the initial conditions (position, momentum) are set to different values so that each trajectory slightly differs from the others. The initial angle and initial velocity are drawn from the same uniform distribution for all data sets. The initial angle ranges from 30° to 170° and the initial velocity ranges from $-2rad/s$ to $2rad/s$. For training, validation and in-distribution testing set, the gravity ranges from $8.0m^2/s$ to $12.0m^2/s$, and the pendulum length ranges from $1.20m$ to $1.40m$. In the easy out-of-distribution testing set, the gravity is between $12.0 - 12.5m^2/s$ and the pendulum length is between $1.40 - 1.45m$, while in the hard out-of-distribution testing set, the gravity is $12.5 - 13.0m^2/s$ and the pendulum length is $1.45 - 1.50m$. The distributions of these parameters are shown in Figure 2.

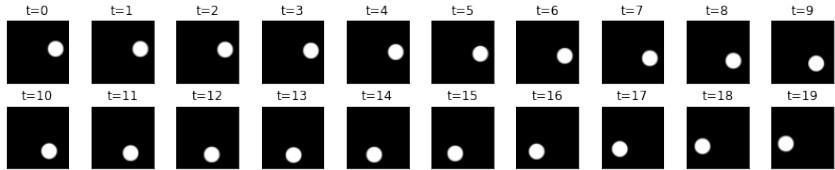

Figure 10: Example image sequence from the video pendulum data set.

# B TRAINING AND HYPERPARAMETERS

## B.1 PHASE SPACE

During training the back-propagation is used after a single forward pass. The input and output of the models are smaller than the sequence size, so to cover the whole sequence we use random starting points per batch, both during training and testing. Both the VAE and MLP AE have an encoder with two hidden layers size 400,200 and a reverse decoder. The LSTM model has two stack LSTM cells with hidden size of 100, which results on an equivalent number of neurons. We used the Adam optimizer with $b_1 = 0.9$ and $b_2 = 0.999$. A scheduler for the learning rate was applied whose patience and scaling factor are hyperparameters. Maximum number of epochs was set to 2000 but we employed also early stopping using a validation set which led to significantly less epochs.

Table 2: **Number experiments with phase space data.** Each experiment corresponds to a distinct configuration of hyperparameters.

|  | MLP | MLP-SD | VAE | VAE-SD | VAE-SSD | LSTM | Total |
|---|---|---|---|---|---|---|---|
| Pendulum | 72 | 72 | 72 | 72 | 72 | 72 | 432 |
| L-V | 72 | 72 | 72 | 72 | 72 | 72 | 432 |
| 3-body | 96 | 96 | 96 | 96 | 96 | 96 | 576 |
|  |  |  |  |  | Total experiments | | **1440** |

Table 3: **Pendulum hyperparameters**

| | MLP | MLP-SD | VAE | VAE-SD | LSTM |
|---|---|---|---|---|---|
| Input Size | | | 10, 50 | | |
| Output Size | | | 1, 10 | | |
| Hidden Layers | | | [400, 200] | | 50,100,200 |
| Latent Size | | | 4, 8, 16 | | - |
| Nonlinearity | | | Leaky ReLU | | Sigmoid |
| Num. Layers | - | - | - | - | 1,2,3 |
| Learning rate | | | $10^{-3}$ | | |
| Batch size | 16, 32 | 16 | 16, 32 | 16 | 16, 64 |
| Sched. patience | 20, 30, 40 | 20,30 | 20 | 20 | 30 |
| Sched. factor | 0.3 | 0.3 | 0.3 | 0.3 | 0.3 |
| Gradient clipping | No | 1.0 | 1.0 | | |
| Layer norm (latent) | No | No | Yes | Yes | No |
| Teacher Forcing | - | - | - | - | Partial |
| Decoder $\gamma$ | - | - | $10^{-3}, 10^{-4}, 10^{-5}$ | $10^{-3}, 10^{-4}$ | - |
| Sup. scaling | - | Linear | - | Linear | - |
| Supervision $\delta$ | - | 0.1, 0.2, 0.3 | - | 0.01, 0.1, 0.2 | - |
| # of experiments | 72 | 72 | 72 | 72 | 72 |

Table 4: **Lotka-Volterra hyperparameters**

| | MLP | MLP-SD | VAE | VAE-SD | LSTM |
|---|---|---|---|---|---|
| Input Size | | | 50 | | |
| Output Size | | | 10 | | |
| Hidden Layers | | | [400, 200] | | 50,100 |
| Latent Size | | | 8, 16, 32 | | - |
| Nonlinearity | | | Leaky ReLU | | Sigmoid |
| Num. Layers | - | - | - | - | 1,2,3 |
| Learning rate | $10^{-3}, 10^{-4}$ | $10^{-3}, 10^{-4}$ | $10^{-3}, 10^{-4}$ | $10^{-3}$ | $10^{-3}$ |
| Batch size | 16, 32, 64 | 16, 32 | 16, 32 | 16 | 10, 64, 128 |
| Sched. patience | 20, 30 | 20, 30 | 20 | 20 | 20, 30 |
| Sched. factor | 0.3, 0.4 | 0.3 | 0.3 | 0.3 | 0.3 |
| Gradient clipping | No | No | 0.1, 1.0 | 0.1, 1.0 | No |
| Layer norm (latent) | No | No | No | No | No |
| Teacher Forcing | - | - | - | - | Partial, No |
| Decoder $\gamma$ | - | - | $10^{-4}, 10^{-5}, 10^{-6}$ | $10^{-4}, 10^{-5}, 10^{-6}$ | - |
| Sup. scaling | - | Linear | - | Linear | - |
| Supervision $\delta$ | - | 0.1, 0.2, 0.3 | - | 0.01, 0.1, 0.2, 0.3 | - |
| # of experiments | 72 | 72 | 72 | 72 | 72 |

Table 5: **3-body system hyperparameters**

| | MLP | MLP-SD | VAE | VAE-SD | LSTM |
|---|---|---|---|---|---|
| Input Size | | | 50 | | |
| Output Size | | | 10 | | |
| Hidden Layers | | | [400, 200] | | 50,100 |
| Latent Size | | | 8, 16, 32 | | - |
| Nonlinearity | | | Leaky ReLU | | Sigmoid |
| Learning rate | $10^{-3}, 10^{-4}$ | $10^{-3}, 10^{-4}$ | $10^{-3}, 10^{-4}$ | $10^{-3}$ | |
| Batch size | 16, 32 | 16 | 16 | 16 | 16, 64, 128 |
| Sched. patience | 30, 40, 50, 60 | 30, 40, 50, 60 | 30, 40, 50, 60 | 30, 40, 50, 60 | 20, 30 |
| Sched. factor | 0.3, 0.4 | 0.3 | 0.3, 0.4 | 0.3, 0.4 | 0.3 |
| Gradient clipping | No | No | No | No | No |
| Layer norm (latent) | No | No | No | No | No |
| Decoder $\gamma$ | - | - | $10^{-5}, 10^{-6}$ | $10^{-5}, 10^{-6}$ | - |
| Sup. scaling | - | Linear | - | Linear | - |
| Supervision $\delta$ | - | 0.05, 0.1, 0.2, 0.3 | - | 0.1, 0.2 | - |
| # of experiments | 96 | 96 | 96 | 96 | 96 |

## B.2 VIDEO PENDULUM

### B.2.1 CNN-VAE MODEL

Encoder has 4 layers convolutional layers with 32, 32, 64 and 64 maps respectively. The filter size is 3, padding is 1 and stride is 2. The last convolutional layer is flattened as a 256-dimensional vector to become the inner VAE input. The decoder 4 convolutional layers (64,64,32,32) with bi-linear upsampling. Model input and out is 20 frames. For the models without supervised disentanglement, a grid search is performed upon the $\beta$ value, the size of the latent space, and the roll-out length during training. For the models with supervised disentanglement, a grid search is performed upon the $\beta$ value, the size of the latent space, the time step of the data set, the roll-out length during training and the supervision multiplier. The detailed search grid is summarised in Table 6. Learning rate was $10^{-3}$ and an Adam optimizer ($b_1 = 0.9$ and $b_2 = 0.999$) was used. We also used early stopping upon the cumulative reconstruction loss for the first 200 steps on a validation set with the max number of epochs being 1000.

Table 6: **Video pendulum hyperparameters for CNN-VAE models**

| | CovnVAE | CovnVAE-SD |
|---|---|---|
| Latent Size | 4, 8, 16 | 8, 16 |
| Decoder $\gamma$ | 0.01, 0.1, 1 | 0.01, 0.1, 1 |
| VAE recursions | 1, 2, 4, 8 | 4, 8 |
| Supervision $\delta$ | - | 0.01, 0.1, 1 |
| # of experiments | 36 | 36 |

### B.2.2 RSSM MODELS

For the RSSM model we follow the architecture parameters as described in Hafner et al. (2018) & Saxena et al. (2021). For training we use sequences of 100 frames and batch size 100. All models were trained for 300 epochs with a learning rate of $10^{-3}$ and an Adam optimizer ($b_1 = 0.9$ and $b_2 = 0.999$). During testing the model uses 50 frames as context (input). The parameters we tune appear in Table 7.

Table 7: **video pendulum hyperparameters for RSSM models**

|  | RSSM | RSSM-SD |
| --- | --- | --- |
| Batch Size | | 50, 100 |
| Decoder std. | | 1.0, 2.0 |
| Train Input Length | | 50, 100 |
| Supervision $\delta$ | - | 0.01, 0.1, 1 |
| Seeds | #3 | #1 |
| # of experiments | 24 | 24 |

## C PHASE SPACE RESULTS

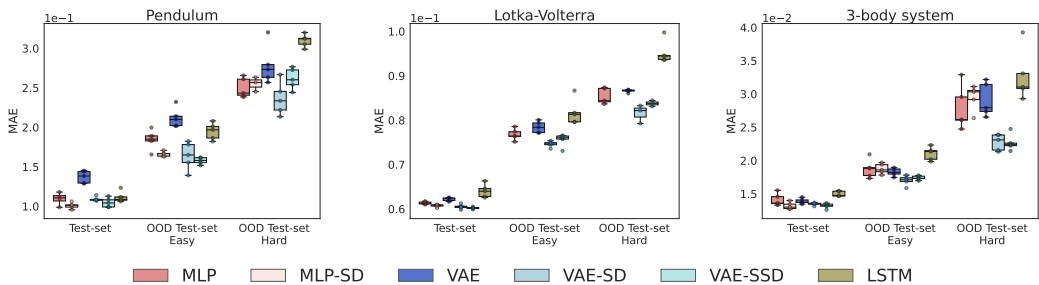

Figure 11: **Mean Absolute Error for first 200 time-steps.** The bars represent the 5 experiments of each model with lower MAE. In all three systems disentangled VAEs provide an advantage over the other baseline. The disentanglement in MLP does not increase performance as consistently. The scaling in VAE-SSD allows it to better capture the parameter space of the original test-set but in most cases VAE-SD generalizes better OOD. Similarly, LSTM does not generalize well OOD.

Table 8: **MAE ($\times 10^2$) of phase space experiments at 200 time-steps.** Average and standard deviation of 5 best models.

|  | Pendulum | | | Lotka-Voltera | | | 3-Body system | | |
| --- | --- | --- | --- | --- | --- | --- | --- | --- | --- |
|  | Test-set | OOD Easy | OOD Hard | Test-set | OOD Easy | OOD Hard | Test-set | OOD Easy | OOD Hard |
| LSTM | 11.16±0.70 | 19.49±1.06 | 30.98±0.80 | 6.41±0.15 | 8.18±0.29 | 9.52±0.26 | 1.50 ± 0.04 | 2.10 ± 0.10 | 3.27±0.39 |
| MLP | 10.96±0.80 | 18.45±1.24 | 24.98±1.25 | 6.13±0.03 | 7.68±0.13 | 8.53±0.18 | 1.41 ± 0.09 | 1.88 ± 0.14 | 2.79±0.33 |
| MLP-SD | **10.08±0.39** | 16.61±0.32 | 25.50±0.92 | 6.07±0.03 | N/A | N/A | **1.32 ± 0.05** | 1.88 ± 0.08 | 2.95±0.19 |
| VAE | 13.71±0.77 | 21.20±1.25 | 27.86±2.48 | 6.22±0.04 | 7.84±0.13 | 8.66±0.03 | 1.39 ± 0.04 | 1.83 ± 0.06 | 2.91±0.25 |
| VAE-SD | 10.92±0.28 | 16.40±1.75 | **23.62±2.09** | 6.05±0.05 | **7.46±0.07** | **8.16±0.21** | 1.35 ± 0.02 | **1.70 ± 0.07** | **2.27±0.12** |
| VAE-SSD | 10.47±0.60 | **15.77±0.42** | 26.15±1.33 | **6.02±0.02** | 7.56±0.14 | 8.38±0.09 | **1.32 ± 0.04** | 1.74 ± 0.03 | **2.27±0.12** |

Table 9: **Models that diverge in at least one trajectory.** Percentage out of the top 5 models selected by validation accuracy.

|  | Pendulum | | | Lotka-Voltera | | | 3-Body system | | |
| --- | --- | --- | --- | --- | --- | --- | --- | --- | --- |
|  | In Dist. | OOD Easy | OOD Hard | In Dist. | OOD Easy | OOD Hard | In Dist. | OOD Easy | OOD Hard |
| LSTM | - | - | - | - | - | - | - | - | - |
| MLP | 20% | - | - | - | - | - | - | - | - |
| MLP-SD | - | - | 40% | - | 100% | 100% | - | - | - |
| VAE | - | - | - | - | - | - | - | - | - |
| VAE-SD | - | - | - | - | - | 40% | - | - | - |
| VAE-SSD | - | - | - | - | - | 60% | - | - | - |

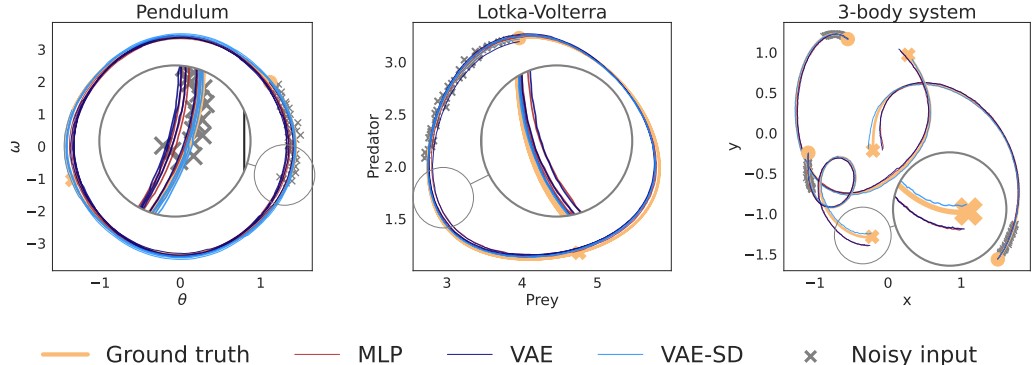

Figure 12: **Model predictions in phase space**. Trajectories are taken from the OOD Test-Set Hard of each system. The model input is noisy. The circle and bold '×' markers denote the start and end of the ground truth trajectories respectively.

## D    VIDEO PENDULUM RESULTS

**LPIPS**

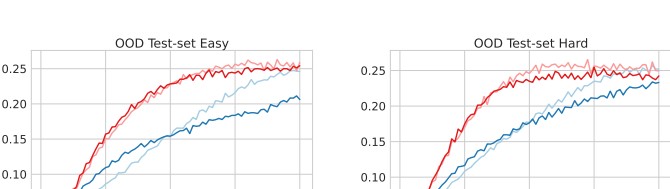

Figure 13: **Prediction quality on the video pendulum** as a function of the distance predicted into the future (x axis). For LPIPS lower is better.

Table 10: **Model comparison in video pendulum.** Metrics are calculated between ground truth and prediction of the models at exactly 800 timesteps in the future.

|  | SSIM | | | LPIPS | | | PSNR | | |
|---|---|---|---|---|---|---|---|---|---|
|  | Test-set | OOD Easy | OOD Hard | Test-set | OOD Easy | OOD Hard | Test-set | OOD Easy | OOD Hard |
| RSSM | 0.795 | 0.787 | 0.783 | 0.227 | 0.246 | 0.252 | 13.36 | 12.71 | 12.26 |
| RSSM-SD | **0.813** | **0.808** | **0.794** | **0.192** | **0.206** | **0.233** | **14.16** | **13.82** | **12.90** |
| CNN-VAE | 0.787 | 0.781 | 0.784 | 0.242 | 0.259 | 0.250 | 12.67 | 12.41 | 12.36 |
| CNN-VAE-SD | **0.789** | **0.783** | **0.788** | **0.241** | **0.254** | **0.242** | **12.76** | **12.46** | **12.53** |

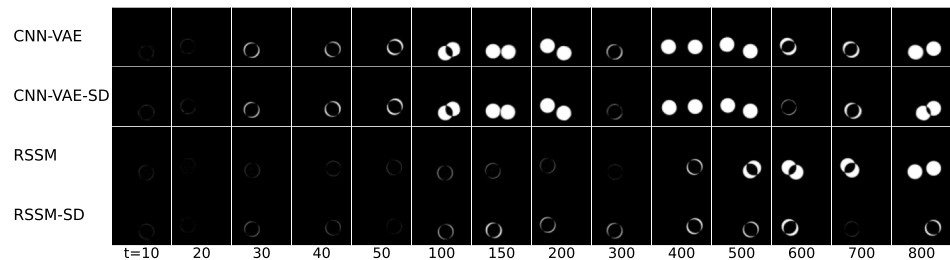

Figure 14: Absolute difference between ground truth and predictions on the OOD Test-set Hard of the video pendulum data set.

