# OpenReview forum: "Disentangled generative models for robust dynamical system prediction"
_ICLR.cc/2022/Conference — ICLR 2022 Submitted_

### Official Review · Reviewer_z3gj · 2021-10-29

**Correctness:** 3
**Technical Novelty And Significance:** 2
**Empirical Novelty And Significance:** 2
**Recommendation:** 1
**Confidence:** 4

**Main Review:**

## Contribution

The tackled problem of out-of-distribution generalization for the forecasting of dynamical systems is relevant and valuable to the community, as motivated in the paper. Models that generalize well in this setting still need to be discovered. This is an important issue as a forecasting system overfitting the training distribution cannot possibly have retained the true dynamics, in this case the ODE / PDE, of the observed phenomenon. The proposed method is one of the first steps in this challenging direction. However, to the best of my understanding, I find the contribution of this paper to be overly limited for acceptance, with numerous limitations that are described below.

### Supervision and Disentanglement

One of the main claims of the paper is that the proposed method makes models disentangle the system's parameters, but this claim is not sufficiently supported. It is clear that the method makes model learn the system's factors of variations by design, but no experiment indicates that it does separate them as well, which is necessary for disentanglement. In this regard, I disagree with the statement at the end of Section 1 removing from the scope of the paper the inspection of the learned representations, given the claims of the paper.

Experiments investigating the latent space could include, for example, the manipulation of the latent variables associated to the systems' parameters to assess whether they are actually disentangled, similarly to e.g. [1, Figure 1]. The advantage of the considered framework of experiments in this setting is that all the data is simulated: one could numerically compare the resulting sequence from the manipulation of the latent state with the actual simulated sequence generated with the corresponding parameters, in line with [2, Sections 5.2 and 5.3] in another context of disentanglement.

Regardless of the disentanglement property, an experiment evaluating the ability of the models learned with the proposed loss to retrieve the true parameters of the system from the observation of the sequence would greatly emphasize the utility of the method. Indeed, estimating the parameters of dynamical systems is an active line of research which this paper directly follows, given that the method is supervised on these parameters.

### Novelty and Significance

The novelty of the proposed method is limited, given prior works on supervised disentanglement in other contexts (e.g. [3] cited in the paper). The discovery that the supervision of the system's parameters improves forecasting performance is interesting but expected; an opposite result may have been questionable. I stress that this is not a significant problem per se, given that evidencing this behavior in the context of dynamical systems is valuable to the community. However, this limited novelty is to be considered with the lack of significance in the presented empirical study.

The lack of significance first lies in the numerical results, which all show non-significant or marginal improvements for the proposed method against the baselines: VAE-SSD performs similarly to VAE-SD, and VAE-SD is only marginally better than VAE and is far from closing the gap between out-of-distribution and in-distribution performance. Furthermore, qualitative animated results that I checked in the Hard OOD setting provided as supplementary do not match the example of Figure 5 as VAE and VAE-SD seem very close compared to the difference between them and the ground truth, making the improvement look thin. To my understanding, these mixed results may be the consequence of choices in experimental design, as argued in the next point of this review.

The experiments also lack significance in their design. The considered out-of-distribution parameter ranges are restricted and close to the training parameters. Without further discussion in the paper to contextualize this choice, it would seem that out-of-distribution sequences might be close to in-distribution sequences, thus questioning the obtained results. I would suggest the authors to either extend the considered ranges of parameters or explain how the current parameters are relevant.

A possible direction to improve this paper in this regard might be to consider a semi-supervised setting, like [3], instead of a fully supervised method like in the current version. Real-world simulations can be expensive to run and the available labeled data in this context may be limited, thus rather motivating a method working with sparsely labeled data.

### Choice of Models

The choice of models to apply the proposed supervision may be questionable and explain the mixed results obtained in the experiments. To the best of my knowledge, forecasting models of the kind of VAE and CNN-VAE in the paper are not widely adopted in the community; I would be interested in references that the authors could provide to support this choice. Instead, state-of-the-art variational models often rely on sequential latent variable generative models, like [4, 5, 6, 7, 8] to name some works in the last five years. Moreover, the use of ODE-like recurrent schemes may be considered as well as they have been shown to be adapted to the prediction of dynamical systems (see e.g. [9, 10, 11]).

RSSM is the only baseline in this paper following this line of works, and is also the only model presenting a substantial improvement with the proposed supervision. I believe that this is no coincidence, given that a questionable choice of model like MLP may induce suboptimal results with the introduced method. I encourage the authors to strengthen their empirical evaluation by considering more robust and standard models.


## Other Remarks and Questions

### Questionable Claims

Several other claims is the paper may be questionable, as listed in the following.

> System identification [...] requires knowledge of the underlying system to be computationally effective". [Page 1]

It would seem that the proposed method does require knowledge of the underlying system as well, since it relies on supervising over the system's parameters.

> [We treat] the ground truth domain parameters from simulations as privileged information which, to the best of our knowledge, has not been applied to dynamical system prediction previously. [Page 2]

This may be a wording issue but privileged information has already been leveraged for dynamical systems for the last few years, cf. for example [10, 11, 12], even though this privileged information is not necessarily the system's parameters. The authors might consider further discussing this point.

> The problem is that [VAEs] usually lack in competitive performance.

Without references to support this claim, I would strongly disagree given the references mentioned above [4, 5, 6, 7, 8].

### Number of Experiments

Figures 3 and 4 are said to show the top 5 models of each architecture, but I could not understand the details of this selection. Does this correspond to the top 5 best performing sets of hyperparameters? Or is it the top 5 over a given number of experiments for the same set of hyperparameters?

### LPIPS

Could the authors justify the choice of LPIPS for the experiments in Section 5? LPIPS is a perceptual metric for realistic images, making it a priori less relevant for synthetic datasets like these pendulum sequences. The authors might rather highlight PSNR which is a standard metric for this type of datasets and is already used in the appendix.

### Writing

The paper is mostly clear and easy to read, but I find the description of the models to be confusing regarding their nature and the considered architectures (for instance, the VAE is underspecified in the main text), which raises issues in the motivation of the modeling choices in the paper as mentioned above. Many figures are hard to read in greyscale; I recommend that the authors improve their readability to make them as accessible as possible.

Typos:
 - the reference to Saxena et al. (2021) at the end of Section 1 should be between parentheses;
 - title of Section 3.2: "disentanglment" should be "disentanglement";
 - title of Section 3.3: there is an extra parenthesis at the end of the title;
 - Section 3.3: "which can be though as" should be "which can be thought of as";
 - there is an extra comma and no parentheses are needed in the last sentence of page 4;
 - Section 4.3: "We also observe that VAE-SSD model the in-distribution data" should be "We also observe that VAE-SSD models the in-distribution data".


## References

[1] I. Higgins et al. $\beta$-VAE: Learning Basic Visual Concepts with a Constrained Variational Framework. ICLR 2017.\
[2] J. Donà et al. PDE-Driven Spatiotemporal Disentanglement. ICLR 2021.\
[3] F. Locatello et al. Disentangling Factors of Variations Using Few Labels. ICLR 2020.\
[4] R. G. Krishnan et al. Structured Inference Networks for Nonlinear State Space Models. AAAI 2017.\
[5] M. Fraccaro et al. A Disentangled Recognition and Nonlinear Dynamics Model for Unsupervised Learning. NIPS 2017.\
[6] E. Denton et al. Stochastic Video Generation with a Learned Prior. ICML 2018.\
[7] J. Chung et al. A Recurrent Latent Variable Model for Sequential Data. NIPS 2015.\
[8] Y. Rubanova et al. Latent Ordinary Differential Equations for Irregularly-Sampled Time Series. NeurIPS 2019.\
[9] R. T. Q. Chen et al. Neural Ordinary Differential Equations. NeurIPS 2018.\
[10] S. Greydanus et al. Hamiltonian Neural Networks. NeurIPS 2019.\
[11] Y. Yin et al. Augmenting Physical Models with Deep Networks for Complex Dynamics Forecasting. ICLR 2021.\
[12] M. Raissi et al. Physics-Informed Neural Networks: A Deep Learning Framework For Solving Forward and Inverse Problems Involving Nonlinear Partial Differential Equations. Journal of Computational Physics. 2019.

**Summary Of The Paper:**

This paper introduces a supervised loss to encourage dynamical systems predictors to retain systems' parameters (e.g. appearing in the underlying ODE) in their latent space. It presents multiple experiments to evaluate the advantages of this approach, including better long-term forecasting ability as well as improved prediction performance for out-of-distribution parameters, that had not been seen during training.

**Summary Of The Review:**

From the limitations that are described in this review, I think that this paper needs very significant changes to be accepted, especially because of the questionable claims and insufficient experimental results. Nonetheless, I am looking forward to discussing my opinion with the authors and other reviewers. I believe that this paper follows an interesting line of research and that further work could make it ready for publication at a next conference.

### Post-Rebuttal Update

I acknowledge the authors' response and thank them for their extensive answer. As explained in my follow-up response, I find that the proposed improvements are marginal and insufficient to raise my score. Therefore, I maintain my strong recommendation to reject the paper.

---

> ### Author Response · Authors · 2021-11-18
> **Comment on Review 5 - A**
>
> We would like to thank you for your review. We are glad you find this line of research interesting. We address specific comments below:
>
> * Supervision and Disentanglement
> One of the main claims of the paper is that the proposed method makes models disentangle the system's parameters, but this claim is not sufficiently supported. It is clear that the method makes model learn the system's factors of variations by design, but no experiment indicates that it does separate them as well, which is necessary for disentanglement. In this regard, I disagree with the statement at the end of Section 1 removing from the scope of the paper the inspection of the learned representations, given the claims of the paper.
> Experiments investigating the latent space could include, for example, the manipulation of the latent variables associated to the systems' parameters to assess whether they are actually disentangled, similarly to e.g. [1, Figure 1]. The advantage of the considered framework of experiments in this setting is that all the data is simulated: one could numerically compare the resulting sequence from the manipulation of the latent state with the actual simulated sequence generated with the corresponding parameters, in line with [2, Sections 5.2 and 5.3] in another context of disentanglement.
> Regardless of the disentanglement property, an experiment evaluating the ability of the models learned with the proposed loss to retrieve the true parameters of the system from the observation of the sequence would greatly emphasize the utility of the method. Indeed, estimating the parameters of dynamical systems is an active line of research which this paper directly follows, given that the method is supervised on these parameters.
> --> We have chosen to prioritized downstream performance in this work. We do think that the downstream task results demonstrate that there is merit to this method, which was our primary objective. We agree that investigating the representations, by adding disentanglement metrics, is important to further enhance our findings. Thank you very much for the suggestions.
>
> * Novelty and Significance
> The lack of significance first lies in the numerical results, which all show non-significant or marginal improvements for the proposed method against the baselines: VAE-SSD performs similarly to VAE-SD, and VAE-SD is only marginally better than VAE and is far from closing the gap between out-of-distribution and in-distribution performance. Furthermore, qualitative animated results that I checked in the Hard OOD setting provided as supplementary do not match the example of Figure 5 as VAE and VAE-SD seem very close compared to the difference between them and the ground truth, making the improvement look thin. To my understanding, these mixed results may be the consequence of choices in experimental design, as argued in the next point of this review.
> --> In Table 8 in the Appendix we present the numerical values of the RMSE. For the OOD datasets, VAE-SD consistently reduces the error in comparison to the VAE by 5-22%.
> For the RSSM (Table 10 in Appendix) there is a 8-16% LPIPS reduction across datasets, 5-9% PSNR increase and approx 1-3% increase in SSIM.
> We would like to give some context why we believe these results are quite important:
> 1. the improvement is against a SoTA dynamical prediction model (RSSM). Improving on it up to 16% in some metrics should not be considered trivial. For example in Saxena et al. (Table 1; NeurIPS 2021), the improvement over RSSM is less than 5% (when there is any).
> 2. they are consistent across datasets
> 3. they come over the whole range of predicted frames (800 timesteps in video pendulum)
> 4. come from quite diverse in and out-of-distribution data-sets.
>
> It is really hard to assess the models qualitatively, the examples only serve to give an intuition of what can go well or wrong in some predictions.

---

> ### Author Response · Authors · 2021-11-18
> **Comment on Review 5 - B**
>
> * The experiments also lack significance in their design. The considered out-of-distribution parameter ranges are restricted and close to the training parameters. Without further discussion in the paper to contextualize this choice, it would seem that out-of-distribution sequences might be close to in-distribution sequences, thus questioning the obtained results. I would suggest the authors to either extend the considered ranges of parameters or explain how the current parameters are relevant.
> --> There is zero overlap between the test-set, OOD Easy and OOD Hard test sets. We had incorrectly explained the tests sets in Table 1 of the Appendix. We have now corrected the Table also added clarifications on Figure 9. We are sorry for the confusion. Overall, we have constructed our dataset so that the training and OOD test-sets are disjoint (Figures 2 & 9 help by visualizing this). Effectively, this guarantees that for all sequences, at least one parameter is outside of the training range. We have also added clarification for the Video Pendulum dataset in the Section A.2 of the appendix, which along with Figure 2, should give a clear picture of the OOD nature of the dataset. It is also important to note that for example in Figure 9 the area of the blue region is about half of the green region (again there is no overlap). This means that the parameter space in the OOD test sets is far from constrained or trivially different. Furthermore, in dynamical systems a small variation in parameters can produce widely different trajectories in the phase space (e.g. bifurcations). Under that light the differences between the datasets are quite substantial.
>
> A possible direction to improve this paper in this regard might be to consider a semi-supervised setting, like [3], instead of a fully supervised method like in the current version. Real-world simulations can be expensive to run and the available labeled data in this context may be limited, thus rather motivating a method working with sparsely labeled data.
> --> Thank you these are great suggestions which we will consider to improve our work.

---

> ### Author Response · Authors · 2021-11-19
> **Comment on Review 5 - C**
>
> * Choice of Models
> The choice of models to apply the proposed supervision may be questionable and explain the mixed results obtained in the experiments. To the best of my knowledge, forecasting models of the kind of VAE and CNN-VAE in the paper are not widely adopted in the community; I would be interested in references that the authors could provide to support this choice. Instead, state-of-the-art variational models often rely on sequential latent variable generative models, like [4, 5, 6, 7, 8] to name some works in the last five years. Moreover, the use of ODE-like recurrent schemes may be considered as well as they have been shown to be adapted to the prediction of dynamical systems (see e.g. [9, 10, 11]). RSSM is the only baseline in this paper following this line of works, and is also the only model presenting a substantial improvement with the proposed supervision. I believe that this is no coincidence, given that a questionable choice of model like MLP may induce suboptimal results with the introduced method. I encourage the authors to strengthen their empirical evaluation by considering more robust and standard models.
> --> There is a growing body of work that uses feed-forward neural networks for spatiotemporal prediction of dynamical systems in physical sciences [1,2,3,4,5]. Models with bottlenecks (U-Nets or similar) are a very common choice and reported results are promising. While VAEs have not been explored a lot, the idea of using it in a feed-forward way for prediction is the same. In hindsight, it is possible that the CNN-VAE models are not adequately powerful and this could explain the limited improvement too. Of course dynamical VAEs is also a very important line of work and this is the reason we include the RSSM model in our experiments which as you mention shows very promising behavior with disentanglement.
> [1] De Bézenac, Emmanuel et al. "Deep learning for physical processes: Incorporating prior scientific knowledge." Journal of Statistical Mechanics: Theory and Experiment (2019)
> [2] Fotiadis, Stathi, et al. "Comparing recurrent and convolutional neural networks for predicting wave propagation." ICLR 2020 Workshop on Integration of Deep Neural Models and Differential Equations. (2020)
> [3] Alguacil, Antonio, et al. "Predicting the propagation of acoustic waves using deep convolutional neural networks." Journal of Sound and Vibration (2021)
> [4] Fresca, Stefania, et al. "POD-Enhanced Deep Learning-Based Reduced Order Models for the Real-Time Simulation of Cardiac Electrophysiology in the Left Atrium." Frontiers in physiology (2021)
> [5] Pinto, Wagner Gonçalves et al.  "On the reproducibility of fully convolutional neural networks for modeling time-space evolving physical systems." arXiv preprint arXiv:2105.05482 (2021)
>
> * System identification [...] requires knowledge of the underlying system to be computationally effective". [Page 1]
> It would seem that the proposed method does require knowledge of the underlying system as well, since it relies on supervising over the system's parameters.
> --> Our assumption is only that there are some factors of variation in the system, we make no assumptions for the governing dynamics like system identification techniques do.
>
> * [We treat] the ground truth domain parameters from simulations as privileged information which, to the best of our knowledge, has not been applied to dynamical system prediction previously. [Page 2]
> This may be a wording issue but privileged information has already been leveraged for dynamical systems for the last few years, cf. for example [10, 11, 12], even though this privileged information is not necessarily the system's parameters. The authors might consider further discussing this point.
> --> By priviledged information we refer to the system parameters being used for supervision. Inductive biases inferred stemming from system knowledge are indeed very common. We have changed the wording in Section 1 to make this clear.
>
> * The problem is that [VAEs] usually lack in competitive performance.
> Without references to support this claim, I would strongly disagree given the references mentioned above [4, 5, 6, 7, 8].
> --> Our understanding is the VAEs considered in the disentanglement setting like beta-VAE and FactorVAE etc (Higgins et al. 2017, Kim & Mnih 2018, Locatello et al. 2019) are not considered up to par with other models generative models. We do not have NLL numbers but these models are absent in recent comparisons for image modelling (Child Rewon; ICRL 2021, Vahdat & Kautz, NeurIPS 2020). Hierarchical and other extensions of VAEs are competitive but we have not seen them being used in the disentanglement setting. In any case we removed the sentence to avoid confusion.

---

> ### Author Response · Authors · 2021-11-19
> **Comment on Review 5 - D**
>
> * Number of Experiments
> Figures 3 and 4 are said to show the top 5 models of each architecture, but I could not understand the details of this selection. Does this correspond to the top 5 best performing sets of hyperparameters? Or is it the top 5 over a given number of experiments for the same set of hyperparameters?
> --> It corresponds to the top 5 best performing sets of hyperparameters, we have ammended the wording on the captions. 72-96 hyperparameter sets have been trained for each model to make this a fair comparison (see Table 2 of the Appendix.)
>
> * LPIPS
> Could the authors justify the choice of LPIPS for the experiments in Section 5? LPIPS is a perceptual metric for realistic images, making it a priori less relevant for synthetic datasets like these pendulum sequences. The authors might rather highlight PSNR which is a standard metric for this type of datasets and is already used in the appendix.
> --> These 3 metrics (LPIPS, SSIM and PSNR) are some of the most commonly used metrics in the literature in video dynamics. Thank you for the suggestion we have moved PSNR to the main text and LPIPS to the appendix.
>
> * Writing
> The paper is mostly clear and easy to read, but I find the description of the models to be confusing regarding their nature and the considered architectures (for instance, the VAE is underspecified in the main text), which raises issues in the motivation of the modeling choices in the paper as mentioned above. Many figures are hard to read in greyscale; I recommend that the authors improve their readability to make them as accessible as possible.
> --> Due to lack of space we give an outline of the VAE and refer the reader to the respective papers for more details (and the appendix). Thank you very much for the suggestions, we tried to make the paper printer-friendly and we will definitely take note of this.
>
> Also thank you for spotting the typos.
>
> We hope that we have addressed at least some of the reviewers concerns, especially we want to reiterate that the datasets are OOD without overlapping. We also hope to have provided some context for the results that will make assessing their significance easier.

---

> ### Comment · Reviewer_z3gj · 2021-11-22
> **Insufficient Improvements**
>
> I would like to thank the authors for their extensive answer. I appreciate the provided clarifications and consequently raise my score of correctness from 2 to 3. Nonetheless, I find that the principal objections that the other reviewers and I raised have remained unanswered or left for future work; see the following points for more details. I encourage the authors to improve their paper in this regard for a future submission, which I believe could lead to a good paper.
>
> ## Disentanglement
>
> I understand that the authors have chosen to leave the inspection of the learned representations for future work. However, I maintain that such an addition is necessary and would greatly strengthen the paper to compensate the lack of novelty of the approach. Moreover, I notice that the updated submission still mentions disentanglement even though no experiment supports this claim.
>
>
> ## Numerical Improvements
>
> I agree that the proposed method does improve the prediction performance. However, the significance of this improvement remains unclear because it lacks context, as also mentioned by the other reviewers. Points of comparison such as weak baselines (Reviewer dcJ9) would help contextualizing this improvement. The difficulty of the task also remains unclear (see the next point), thereby further questioning the significance of the experimental results.
>
> Regarding the work of Saxena et al., I believe that the comparison may be biased because their work mainly deals with video prediction, which may not be directly comparable to synthetic dynamical systems prediction.
>
>
> ## OOD Setting
>
> I did understand in my first review of the paper that the training and OOD testing sets are disjoint. The small discussion introduced in the updated submissions certainly helps understanding the difficulty of the task but rigorously assessing this difficulty requires more work: to what extent do the sequences deviate from each other with slight changes of parameters for each dataset? Furthermore, the performance drop of state-of-the-art models between in-distribution and OOD testing sets could be a good proxy to evaluate the difficulty of the task (see the next point).
>
>
> ## Choice of Models
>
> I disagree with authors regarding the state-of-the-art nature of the investigated models besides RSSM. Firstly, no ODE-based architecture has been considered in the experiments. Secondly, to the best of my understanding, feedforward networks are indeed used for the prediction of dynamical systems but in a recurrent manner by recurrently predicting the next observation only; in the paper, the feedforward networks are used to predict multiple future steps simultaneously, which is not standard to my knowledge. I would suggest the authors to employ state-of-the-art sequential VAEs like RSSM in most or all of their experiments as it would better support their claims.

---

### Official Review · Reviewer_2NUz · 2021-10-29

**Correctness:** 3
**Technical Novelty And Significance:** 2
**Empirical Novelty And Significance:** 2
**Recommendation:** 3
**Confidence:** 3

**Main Review:**

- The novelty of this work may be not enough for ICLR acceptance standards in that the authors applied existing VAEs with minor modifications to known problems.
- I think the comparison with unsupervised disentanglement models is quite unfair because the proposed model is trained using strong supervision about data. Furthermore, the results showing that supervised disentanglement methods outperform unsupervised ones are trivial and not particularly impressive. Instead of the baselines designed by the authors, it would be better to add some comparisons with existing papers for dynamical system prediction (particularly in Figures 3 and 4).
- I am not sure whether the performance differences in Figures 3 and 4 are statistically significant because the results of some models exhibit quite high variances and the number of examined models (i.e., 5) seems small. It would be better to conduct some statistical tests to show that the differences are meaningful.
- The experiments were conducted only on simple simulated datasets. I think some experiments on real-world and/or more complex data are necessary to show the applicability of the proposed method.
- I think a deeper analysis on disentangle representations is not out-of-scope and is necessary because the paper heavily relied on VAEs for disentanglement learning. (i) Are the latent dimensions obtained with the supervision (z_1:k) truly disentangled? (ii) What kind of information is encoded in the other features without the supervision (z_k+1:d)? It would be better to add quantitative results based on existing disentanglement metrics and/or visual results (latent traversal, embedding space visualization).
- Regarding Figure 7, it would be better to add proper explanation about why RSSM is better for the initial timesteps than RSSM-SD.
- It would be better to improve the presentation quality of Figure 5. It is difficult to identify the differences between the lines in the current version because they are largely overlapped. Simply changing linear axis scales into log scales may be helpful.
- Is the reconstruction loss in page 4 replaced by the prediction loss as described in the caption of Figure 1? If so, please modify the reconstruction loss in page 4 to accurately show the prediction loss.


**Summary Of The Paper:**

The paper introduces a VAE-based disentanglement model for dynamical system prediction, which was trained under the supervision using domain parameters. The authors conducted experiments on simulated datasets and showed good performance for OOD cases and long-term predictions.

**Summary Of The Review:**

I think the underlying technical contributions are quite small, while the empirical results are not particularly impressive. I thus find it difficult to argue for acceptance of the work.

---

> ### Author Response · Authors · 2021-11-18
> **Comment on Review 4**
>
> We would like to thank you for your review. We address specific comments below:
>
> The novelty of this work may be not enough for ICLR acceptance standards in that the authors applied existing VAEs with minor modifications to known problems.
> --> Our main focus was not to provide of a new model but try existing methods like disentanglement in VAEs in a new setting (dynamical system prediction). Furthermore we thoroughly test these methods in OOD settings, which is something that previous works have been missing.
>
> I think the comparison with unsupervised disentanglement models is quite unfair because the proposed model is trained using strong supervision about data. Furthermore, the results showing that supervised disentanglement methods outperform unsupervised ones are trivial and not particularly impressive. Instead of the baselines designed by the authors, it would be better to add some comparisons with existing papers for dynamical system prediction (particularly in Figures 3 and 4).
> --> Our main goal was not to necesarilly produce state-of-the-art results, but to test wether supervised disentanglement can be helpful. We demonstrated this for a simple VAE in the phase space experiments. Nevertheless in the video pendulum experiments we improve upon a SoTa model.
> In Table 8 in the Appendix we present the numerical values of the RMSE. For the OOD datasets, VAE-SD consistently reduces the error in comparison to the VAE by 5-22%.
> For the RSSM (Table 10 in Appendix) there is a 8-16% LPIPS reduction across datasets, 5-9% PSNR increase and approx 1-3% increase in SSIM.
> We would like to give some context why we believe these results are quite important:
> 1. the improvement is against a SoTA dynamical prediction model (RSSM). Improving on it up to 16% in some metrics should not be considered trivial. For example in Saxena et al. (Table 1; NeurIPS 2021), the improvement over RSSM is less than 5% (when there is any).
> 2. they are consistent across datasets
> 3. they come over the whole range of predicted frames (800 timesteps in video pendulum)
> 4. come from quite diverse in and out-of-distribution data-sets.
>
> I am not sure whether the performance differences in Figures 3 and 4 are statistically significant because the results of some models exhibit quite high variances and the number of examined models (i.e., 5) seems small. It would be better to conduct some statistical tests to show that the differences are meaningful.
> --> We chose to present the top 5 models out of 72-96 total trained models. We didn't chose more because that would only increase the variance (qualitatively the results are the same).
>
> The experiments were conducted only on simple simulated datasets. I think some experiments on real-world and/or more complex data are necessary to show the applicability of the proposed method.
> --> We agree this is very important and we consider it for future work.
>
> * I think a deeper analysis on disentangle representations is not out-of-scope and is necessary because the paper heavily relied on VAEs for disentanglement learning. (i) Are the latent dimensions obtained with the supervision (z_1:k) truly disentangled? (ii) What kind of information is encoded in the other features without the supervision (z_k+1:d)? It would be better to add quantitative results based on existing disentanglement metrics and/or visual results (latent traversal, embedding space visualization).
> --> Thank you for your suggestion, we take note of this since it has been mentioned by other reviewers too.
>
> * Regarding Figure 7, it would be better to add proper explanation about why RSSM is better for the initial timesteps than RSSM-SD.
> --> We believe this can be a trade-off between short-term accuracy vs long-term and OOD accuracy, due to disentanglement acting as a regularization. Nevertheless, when accounting for cummulative metrics, RSSM-SD is better in all metrics.
>
> * Is the reconstruction loss in page 4 replaced by the prediction loss as described in the caption of Figure 1? If so, please modify the reconstruction loss in page 4 to accurately show the prediction loss.
> --> Yes, thank you, this have been rectified in the updated manuscript.

---

> > ### Comment · Reviewer_2NUz · 2021-11-21
> > **After the rebuttal**
> >
> > I appreciate the authors' response. Overall, I think the paper has potential, but I am still concerned about the limited novelty and weak baselines. Therefore, I would like to maintain my score.

---

### Official Review · Reviewer_dcJ9 · 2021-11-01

**Correctness:** 3
**Technical Novelty And Significance:** 2
**Empirical Novelty And Significance:** 2
**Recommendation:** 6
**Confidence:** 4

**Main Review:**

Pros:
- A relevant problem in dynamical VAEs that is sufficiently motivated in the paper.
- Empirical experiments on three problems: LV, video pendulum and the three-body problem. Demonstrate long term trajectory prediction and OOD on an easy and a hard task.
- Have done a hyperparameter search and presented some ablation studies.

I find it is an interesting work. However, I strongly feel the authors have not adequately demonstrated the benefit of disentanglement and are missing comparison. My main concerns are below:

-  Lack of evidence on whether supervised loss disentangles dynamic parameters from domain parameters. The authors mention evaluation of disentanglement is beyond the scope. I beg to differ for two reasons:
     - the long term trajectory prediction doesn’t necessarily benefit from the disentanglement of latent factors. There are several methods that achieve good performance in long-term trajectory prediction without any explicit form of disentanglement, for example, Hamiltonian Neural Networks, Hamiltonian generative network (HGN), Symplectic RNN [4], Physics-as-Inverse-Graphics [7], Lagrangian Neural Networks [6], etc.  In the related work section, the authors refer to HNNs and say disentanglement is not successfully addressed in such models. It would help if the authors could elaborate the sentence here. The HNNs learn Hamiltonians in a data-driven way and can make long term predictions. So why do they need to address disentanglement in the first place? It is unclear what added gain comes from supervised disentanglement and how it is advantageous over other state-of-the-art methods of long term trajectory predictions or OOD generalisation.

     - Can supervised loss ensure the domain components are fully disentangled from dynamics? I think it is critical to demonstrate whether domain variables are disentangled from the dynamics in any meaningful way? Could, for instance, fix the domain variables and draw samples by slowly changing the dynamic variable and vice-versa. Or better report disentanglement metrics.

- A weak baseline. As referenced above, several works on extending VAEs to dynamical models have shown empirical and theoretical (symplectic structure) arguments for long-term trajectory prediction. It would be worth comparing those methods and demonstrating any benefit in using a supervised approach. In addition, models like SINDY [5] can discover dynamical parameters in an unsupervised way and have demonstrated benefits on long-term trajectory prediction. Without comparison, it is not evident what the immediate benefits of the supervised setup are? If it is OOD generalisation, authors should at least show this as a limitation in existing approaches.

- It is not apparent what makes the interval of domain parameters an easy or hard problem. It would be beneficial to discuss from the dynamical system perspective.

- The choice of loss in supervised disentanglement needs more explanation. In Section 3.2, it is L1 and in Section 3.3 is L2.

- In Table 9, the results of VAE-SD and VAE-SSD are unstable in some cases. But this is not the case with LSTM or VAE.  The authors should provide some discussion here and a potential explanation of the effect.

Minor comments:

- In Table 1, the domain parameters of the train/val/test set are in the same range. It is likely for a model to perform well on val/test if it has seen sequences of the same parameters in training. Shouldn’t the two be selected differently?
- Please number the equations.

Technical inconsistencies:

-In Section 3.2, the loss function is inconsistent with Figure 1. According to Figure 1, the input to VAE is x_n, and the prediction is x_{n+1}. The loss is a typical VAE plus a supervised disentangled term. There are no dynamics there. If the reconstruction term is supposed to be a prediction of x_{n+1} please add appropriate suffixes on x or z in $\mu_x(z;\theta)$. If this is not the case, please provide details on how dynamics are taken into account.

- Please use consistent scripts. In Section 3.2, the k components of latent variable z are written as under script z_{1:k} and in Section 3.3 for latent variables as superscripts s^{1:k}.

- In loss formulation of Section 3.2, the domain parameters $\xi^{i}$ are associated with sample $x^{i}$. As far as I understand, the time steps in a sequence share the domain parameters. It would be helpful to use a suitable script to express it consistently.

- In the loss formulation of Section 3.3, the prediction model is in the state space. The domain parameters are shared over T; why use the prediction model on $s_t$ instead of $d-k$ components of $s_t$?

# References:

[1] Chang MB, Ullman T, Torralba A, Tenenbaum JB. A compositional object-based approach to learning physical dynamics.

[2] Sanchez-Gonzalez A, Bapst V, Cranmer K, Battaglia P. Hamiltonian graph networks with ode integrators.

[3] Toth P, Rezende DJ, Jaegle A, Racanière S, Botev A, Higgins I. Hamiltonian generative networks.

[4] Chen Z, Zhang J, Arjovsky M, Bottou L. Symplectic recurrent neural networks.

[5] Champion K, Lusch B, Kutz JN, Brunton SL. Data-driven discovery of coordinates and governing equations.

[6] Cranmer M, Greydanus S, Hoyer S, Battaglia P, Spergel D, Ho S. Lagrangian neural networks.

[7] Jaques M, Burke M, Hospedales T. Physics-as-inverse-graphics: Joint unsupervised learning of objects and physics from video.


**Summary Of The Paper:**

In this paper, the authors propose a supervised approach to disentangle domain parameters from the dynamics in a deep latent variable model like VAE. Extending VAEs to dynamical systems is a relevant problem and has been a focus of interest in many recent works [1,2,3,4,5,6,7]. This paper identifies two issues for developing dynamical VAEs,

i) out of distribution generalisation

ii) long term trajectory prediction

The main contribution is to address the aforementioned issues using a supervised loss defined between latent variables and domain parameters. The authors present empirical experiments to support the idea.

**Summary Of The Review:**

In my view, this paper proposes a fair approach to a relevant problem. However, there are several concerns.
- The benefit of disentanglement is not demonstrated. The long term generation is not sufficient to support the claim. If a fully unsupervised approach can work equally good what is the incentive of supervised loss? Therefore, I think it is critical to compare with some of the methods outlined above.
- The contribution is marginal as it simply introduces a regularisation term and provides empirical results. Simplicity is generally good and not a downside. However, it should be supported by proper justification and if possible perhaps by a theoretical claim. The choice of L1 in 3.2 and L2 in 3.3 is not properly explained.
- There are technical inconsistencies that leave room for ambiguities.

I have come to the conclusion this paper has concerns that need addressing. I, therefore, give a score of 5.

# Post Rebuttal

I have changed my score from 5 to 6.

---

> ### Author Response · Authors · 2021-11-18
> **Comment on Review 3 - A**
>
> We would like to thank you for your review. We are glad that you found our work to be  interesting. We address specific comments below:
>
> * There are several methods that achieve good performance in long-term trajectory prediction without any explicit form of disentanglement, for example, Hamiltonian Neural Networks, Hamiltonian generative network (HGN), Symplectic RNN [4], Physics-as-Inverse-Graphics [7], Lagrangian Neural Networks [6], etc. In the related work section, the authors refer to HNNs and say disentanglement is not successfully addressed in such models. It would help if the authors could elaborate the sentence here. The HNNs learn Hamiltonians in a data-driven way and can make long term predictions. So why do they need to address disentanglement in the first place? It is unclear what added gain comes from supervised disentanglement and how it is advantageous over other state-of-the-art methods of long term trajectory predictions or OOD generalisation.
> --> In Hamiltonian Neural Network and Langrangian Neural Networks the domain is fixed and only variability on the initial conditions is assumed. Hence OOD generalization in new domains is not even considered. Barber et al. (2021) takes into account different domains but still does not assess OOD generalization. We have added these clarifications to the text.
>
> * Can supervised loss ensure the domain components are fully disentangled from dynamics? I think it is critical to demonstrate whether domain variables are disentangled from the dynamics in any meaningful way? Could, for instance, fix the domain variables and draw samples by slowly changing the dynamic variable and vice-versa. Or better report disentanglement metrics.
> --> We have chosen to prioritized downstream performance in this work. We do think that the downstream task results demonstrate that there is merit to this method, which was our primary objective. Thank you for the suggestion, we agree that investigating the representations, by adding disentanglement metrics, is important to further enhance our findings.
>
> * A weak baseline. As referenced above, several works on extending VAEs to dynamical models have shown empirical and theoretical (symplectic structure) arguments for long-term trajectory prediction. It would be worth comparing those methods and demonstrating any benefit in using a supervised approach. In addition, models like SINDY [5] can discover dynamical parameters in an unsupervised way and have demonstrated benefits on long-term trajectory prediction. Without comparison, it is not evident what the immediate benefits of the supervised setup are? If it is OOD generalisation, authors should at least show this as a limitation in existing approaches.
> --> We compare the VAE-SD to the normal VAEs and RSSM to RSSM-SD show their comparative strengths and weaknesses. Many current approaches for modelling dynamical systems (i.e. Hamiltonian or Langrangian Neural Networks) use a fixed domain for training and testing. This is very restrictive. While others use a range of domains they do not measure the OOD generalization in depth (Miladinovic et al.) or not at all (Barber et all). There are a lot of open questions regarding OOD generalization in dynamical systems prediction and our work hopefully provides some insight.
>
> * It is not apparent what makes the interval of domain parameters an easy or hard problem. It would be beneficial to discuss from the dynamical system perspective.
> --> Different parameters can produce widely different trajectories in the phase space. This is more exagerated in systems that can exhibit chaotic behavior like the 3-body system but it is also present in less complex systems. A motivating example can be bifurcations. Bifurcations occur when small change in the parameters of a system cause a sudden qualitative change in its behaviour. We added a short summary of this in the introduction.
>
> * The choice of loss in supervised disentanglement needs more explanation. In Section 3.2, it is L1 and in Section 3.3 is L2.
> --> These were empirical choices, we have clarified this in both paragraphs.
>
> * In Table 9, the results of VAE-SD and VAE-SSD are unstable in some cases. But this is not the case with LSTM or VAE. The authors should provide some discussion here and a potential explanation of the effect.
> --> To be fair, all models exhibited some form of instability (the table reports only the top 5 models). The problem with VAE-SD is in the L-V system we hypothesize that one reason is that variability in trajectories in this dataset. Some VAE-SD instances indeed seem more unstable in the L-V system which might mean that more care is needed in tuning, training and selection these models.

---

> ### Author Response · Authors · 2021-11-18
> **Comment on Review 3 - B**
>
> * In Table 1, the domain parameters of the train/val/test set are in the same range. It is likely for a model to perform well on val/test if it has seen sequences of the same parameters in training. Shouldn’t the two be selected differently?
> --> There is zero overlap between the test-set, OOD Easy and OOD Hard test sets. We had incorrectly explained the tests sets in Table 1 of the Appendix. We have now corrected the Table also added clarifications on Figure 9. We are sorry for the confusion. Overall, we have constructed our dataset so that the training and OOD test-sets are disjoint (Figures 2 & 9 help by visualizing this). Effectively, this guarantees that for all sequences, at least one parameter is outside of the training range. We have also added clarification for the Video Pendulum dataset in the Section A.2 of the appendix, which along with Figure 2, should give a clear picture of the OOD nature of the dataset. It is also important to note that for example in Figure 9 the area of the blue region is about half of the green region (again there is no overlap). This means that the parameter space in the OOD test sets is far from constrained or trivially different. Furthermore, in dynamical systems a small variation in parameters can produce widely different trajectories in the phase space (e.g. bifurcations). Under that light the differences between the datasets are quite substantial.
>
> * Please number the equations.
> --> We have numbered the equations in the updated manuscript.
>
> * Please use consistent scripts. In Section 3.2, the k components of latent variable z are written as under script z_{1:k} and in Section 3.3 for latent variables as superscripts s^{1:k}. In loss formulation of Section 3.2, the domain parameters  are associated with sample . As far as I understand, the time steps in a sequence share the domain parameters. It would be helpful to use a suitable script to express it consistently.
> --> Thank you, we have updated the notation to be consistent with Figure 1.
>
> In the loss formulation of Section 3.3, the prediction model is in the state space. The domain parameters are shared over T; why use the prediction model on  instead of  components of ?
> -->  The decoder predicts using both the latent dynamics and the parameters. As we mention in the conclusions, the motivation for this is to be closer to how numerical solvers work and our assumption is that the prediction will be more accurate this way. This is a line of reasoning that other models like Hamiltonian Neural Networks have followed.

---

> > ### Comment · Reviewer_dcJ9 · 2021-11-21
> > **Post Rebuttal**
> >
> > I want to thank the authors for a detailed response. I have gone through the paper again. I still find the benefits of the methods require more justification. The claim on disentanglement is still limited. Methods like SINDy can discover dynamics parameters in an unsupervised way and also generalise to unseen dynamics parameters. The empirical results don't show a significant or consistent improvement on the OOD task. I, therefore, don't see long term generation as a sufficient evaluation criterion.
> >
> > I see the authors have added a sentence on the role of dynamical parameters. I find this discussion crucial for the paper's central claim on OOD. It would be better to demonstrate this empirically, perhaps as a limitation in baseline comparisons.
> >
> > I still see technical inconsistencies in the paper. In section 3.2, the reconstruction term is due to Laplace distribution, but in section 3.3, this is not specified. Also, the parameters of mu_z and sigma_z are typically different. There is an encoder network 'phi' and two linear transformations to get mu_z and logsigma_z. In Equation(3) mu_z and sigma_z have the same parameter 'phi' that fails to capture two linear transformations.
> >
> > Please use consistent math notations. In (1) log and in (4) it is ln.
> >
> > Overall I find the paper in its current form is still limited and lacking novelty. Given some of the clarification provided by the authors I have changed my score from 5 to 6.

---

### Official Review · Reviewer_yFXa · 2021-11-02

**Correctness:** 3
**Technical Novelty And Significance:** 3
**Empirical Novelty And Significance:** 2
**Recommendation:** 5
**Confidence:** 3

**Main Review:**

**Strengths:**
* Clear statement of the underlying hypothesis being tested
* Clear presentation of the results and supporting information
* Extensive sweeps over hyperparameters

**Weaknesses:**
* Improvement of the models with disentanglement in the phase space setting appear marginal; Based on the provided visualizations it is not clear that there is a systematic way in which models with disentanglement perform better. A more expressive analysis of the errors might be helpful to assess this aspect (maybe distribution of errors across the dataset for several fixed samples?)
* It’s hard to assess how much variance in the performance is present in the video prediction metric; This is general challenge with selecting best performing models, as they completely mask away the error bars; (Providing several model instances would help to evaluate the significance better)
* While marginal improvements are presented in coarse performance metrics, an insight into the type/class of errors that are being reduced would be very interesting.
* One potentially important hyperparameter (time step) was not varied, which often significantly affects the prediction accuracy.


**Summary Of The Paper:**

The paper studies the performance of dynamical systems learned from data with a focus on out of distribution (OOD) evaluations. Authors consider the question whether disentangling dynamical system parameters in the latent space can improve the generalization of the models, which is perceived as privileged information available from the reference (ground truth) simulations. Authors carry out experiments on several dynamical systems: pendulum, Lotka-Volterra system and three-body problem. Additionally an experiment on video prediction of a singing pendulum is performed. Authors found that additional disentanglement can improve generalization performance of the models and in video prediction setting leads to better long-term predictions based on structural and perceptual image metrics.

**Summary Of The Review:**

Authors present a clear investigation of how disentanglement of the domain factors may affect the performance of learned dynamical models. The suggested experimental evaluation is sound, but current results seem a little marginal. With additional results/modifications I believe this work could be useful to a wider audience, but my initial rating is marginally below the threshold.

---

> ### Author Response · Authors · 2021-11-18
> **Comments on Review 2**
>
> We would like to thank you for your review. We address specific comments below:
>
> * Improvement of the models with disentanglement in the phase space setting appear marginal; Based on the provided visualizations it is not clear that there is a systematic way in which models with disentanglement perform better. A more expressive analysis of the errors might be helpful to assess this aspect (maybe distribution of errors across the dataset for several fixed samples?)
> --> In Table 8 in the Appendix we present the numerical values of the RMSE. For the OOD datasets, VAE-SD consistently reduces the error in comparison to the VAE by 5-22%.
> For the RSSM (Table 10 in Appendix) there is a 8-16% LPIPS reduction across datasets, 5-9% PSNR increase and approx 1-3% increase in SSIM.
> We would like to give some context why we believe these results are quite important:
> 1. the improvement is against a SoTA dynamical prediction model (RSSM). Improving on it up to 16% in some metrics should not be considered trivial. For example in Saxena et al. (Table 1; NeurIPS 2021), the improvement over RSSM is less than 5% (when there is any).
> 2. they are consistent across datasets
> 3. they come over the whole range of predicted frames (800 timesteps in video pendulum)
> 4. come from quite diverse in and out-of-distribution data-sets.
>
> * It’s hard to assess how much variance in the performance is present in the video prediction metric; This is general challenge with selecting best performing models, as they completely mask away the error bars; (Providing several model instances would help to evaluate the significance better)
> --> We acknowledge that training multiple model instances would provide a better idea of the error spread. For what it's worth, we have trained various models (with different hparams) and we found that the behaviour of the top 3-5 disentangled and non-disentangled models was, on average, qualitatively similar to the best model. We trained 120 models in total, so this should be a good rough estimate for the error spread.
>
> * While marginal improvements are presented in coarse performance metrics, an insight into the type/class of errors that are being reduced would be very interesting.
> --> Analysis of errors is definitely an important next step which we will consider for future work.
>
> * One potentially important hyperparameter (time step) was not varied, which often significantly affects the prediction accuracy.
> --> In initial experiments we tried different time steps, which is indeed a very interesting parameter. We decided to leave it out of the scope of this work. To facilitate comparisons, we assume that even in the OOD case the timestep/framerate would still be the same.

---

> > ### Comment · Reviewer_yFXa · 2021-11-29
> > **Response to the authors**
> >
> > I would like to thank the authors for updating the manuscript and their response to the comments.
> >
> > After reviewing again the results in the current manuscript, I'm still worried that the claim of consistent improvement is well supported by the experimental results. The averaged RMSE error can be quite sensitive to the time window selected for evaluation (later times contribute much more to the RMSE since the deviation is larger, which also increases the sensitivity to the random seed used during training).
> >
> > As a thought for better comparison of the performance one could consider error vs simulation time analysis on an ensemble of models (different random seeds).
> >
> > I think the paper made a step closer to being ready for publication, but not quite there yet in my opinion. I think including additional error analysis and a random seed samples would be great next steps.

---

### Official Review · Reviewer_HYc5 · 2021-11-03

**Correctness:** 2
**Technical Novelty And Significance:** 2
**Empirical Novelty And Significance:** 2
**Recommendation:** 3
**Confidence:** 4

**Main Review:**

- My main issue is the limited novelty of the proposed method. This method is a straightforward extension of the unsupervised disentangled state-space model (Miladinovic et al.) to a supervised one, where the privileged information regarding domain parameters is explicitly fed to the model.
- Certain claims (e.g. regarding disentanglement) are made without proper quantitative and/or qualitative investigation(s).
- It is claimed that the proposed supervised disentanglement method improves performance over the unsupervised method. However, there are no comparisons with the closest unsupervised method  (e.g. Miladinovic et al.). Therefore it is hard to judge whether the proposed supervised method truly performs better or not.
- The results on OOD generalization can not be considered OOD as the parameters' range used to create OOD datasets highly overlap with the ranges of the training datasets (Table 1, Appendix).

I expand on these points in the following:

- Regarding line 1 in the contributions section: The treatment of domain parameters as factors of variation is one of the main proposals of DSSM in Miladinovic et al. DSSM seeks to disentangle these true parameters (also referred to as domain-invariant state dynamics) from observations only. Therefore, I believe it's not the first work to consider this setting.
- The main contribution of this work is the supervised disentanglement of sequential data. However, the authors do not investigate how good the disentanglement is. This leaves room for interpretation i.e. whether it is truly disentanglement that is helping the model in achieving good performance. I believe both quantitative and qualitative analysis of disentanglement would further strengthen the claims made in this paper.
- The true parameters (factors of variation) are explicitly provided to the network for training. The method is not directly comparable to Locatello et al. 2019, as only a few labels were used in Locatello et al. 2019 which resulted in a semi-supervised disentanglement setting, in contrast to a fully supervised setting in this work.
- The authors claim that the supervised disentanglement of the sequential model is better than the unsupervised disentanglement done in DSSM  (Miladinovic et al.). I would appreciate it if the authors could back it up with some empirical evidence. It is important to compare results with DSSM (even Kalman VAE) to see the true benefits of supervision.
- The method is practically limited as the privileged true parameter information is not readily available in real-world systems. Thus, as acknowledged by the authors, this method can only work for the simulated systems where these variables are known beforehand.
- The ranges of the parameters used to create the OOD dataset highly overlaps with the ranges used to create the training dataset. In my opinion, this is not OOD as it is very likely that the test sample comes from the range which is used for training. I suggest authors use the ranges which are completely outside the ranges of the training distribution i.e. extrapolation generalization regime (or even interpolation regime where the parameters are sampled from the subset of the training range but that subset range is not seen during training).
- Have the authors tried Gaussian distribution (correspondingly L2 loss) for the decoder? I wonder how the results might differ from the Laplace distribution.
- The prediction quality is reported by using perceptual metrics LPIPS and SSIM. These metrics compare the deep feature space and statistical properties of the images respectively. I think these metrics are not sufficient for evaluating the predictions of the dynamic. If it is possible then kindly report RMSE and/or NLL.
- Fig1 caption: Do the input-output dimensions differ? I don't think the labeling in the figure is correct as there are some inconsistencies. For e.g. in a single time step: $x_1$ → $x_{n+1}$ and $x_n$ → $x_{n+o}$.

Typos (minor):

pg2: "This can be extend to"

pg2:  "high-dimemnsional video rendering"

pg2:  The sentence "we directly assess using the downstream prediction task." seems incomplete.

pg4: "though as equivalent to the phase space of the system."

pg4: check sentence structure of "being a state-of-the-art model in long-term video prediction,"

pg5: "on three well studies dynamical systems,"

pg9: "prediction is based both on them which"

**Summary Of The Paper:**

This paper introduces a supervised disentanglement method to learn dynamical systems. The method relies on the provision of privileged information (true parameters of a sequence) in order to disentangle them from observations. The method is evaluated on three toy datasets.

**Summary Of The Review:**

I have some reservations regarding the novelty of this work. Moreover, certain empirical results do not back the claims made in this paper. Therefore, I am inclined towards rejection.

---

> ### Author Response · Authors · 2021-11-18
> **Comments on Review 1 - A**
>
> First of all thank you very much for your detailed feedback. We address specific comments below:
>
> * Regarding line 1 in the contributions section: The treatment of domain parameters as factors of variation is one of the main proposals of DSSM in Miladinovic et al. DSSM seeks to disentangle these true parameters (also referred to as domain-invariant state dynamics) from observations only. Therefore, I believe it's not the first work to consider this setting.
> --> Domain parameters as factors of variation have been considered in the past (including Miladinovic et al. (2019) and Li et al. (2018)), our difference lies in using the priviledged information during training, we make changes to the wording in Section 1, to more clearly reflect that. Furthermore, previous works, including Miladinovic et al., assess OOD performance in a very limited way. We believe that our experiments give a more rounded perspective into OOD generalization, especially how models fail and what can be done to fix it.
>
> * The main contribution of this work is the supervised disentanglement of sequential data. However, the authors do not investigate how good the disentanglement is. This leaves room for interpretation i.e. whether it is truly disentanglement that is helping the model in achieving good performance. I believe both quantitative and qualitative analysis of disentanglement would further strengthen the claims made in this paper.
> --> We have chosen to prioritize downstream performance in this work. We do think that the downstream task results demonstrate that there is merit to this method, which was our primary objective. Thank you for the suggestion, we agree that investigating the representations, by adding disentanglement metrics, is important to further enhance our findings.
>
> * The authors claim that the supervised disentanglement of the sequential model is better than the unsupervised disentanglement done in DSSM (Miladinovic et al.). I would appreciate it if the authors could back it up with some empirical evidence. It is important to compare results with DSSM (even Kalman VAE) to see the true benefits of supervision.
> --> We compare the VAE-SD to the normal VAEs, we could not find a statement in our paper about VAE-SD being better than DSSM. If the reviewer could kindly point to it we would be happy to change to the wording to make this more clear. While, we do agree that a comparison with DDSM could be beneficial, our main goal was not to necesarilly produce state-of-the-art results, but to test wether supervised disentanglement can be helpful. We demonstrated this for a simple VAE in the phase space experiments. In the video experiments we extended this to a powerful, SoTA, model (RSSM). We understand your comment that an analogous effort on phase space data, is much justified.
>
> * The method is practically limited as the privileged true parameter information is not readily available in real-world systems. Thus, as acknowledged by the authors, this method can only work for the simulated systems where these variables are known beforehand.
> --> Indeed, we acknowledge in the paper this limitation of the method. Nevertheless, our work can motivate future or parallel work in a semi-supervised setting. Also, as the field of transfering models trained in simulation to real data is maturing,the importance of this limitation lessens.

---

> ### Author Response · Authors · 2021-11-18
> **Comments on Review 1 - B**
>
> * The ranges of the parameters used to create the OOD dataset highly overlaps with the ranges used to create the training dataset. In my opinion, this is not OOD as it is very likely that the test sample comes from the range which is used for training. I suggest authors use the ranges which are completely outside the ranges of the training distribution i.e. extrapolation generalization regime (or even interpolation regime where the parameters are sampled from the subset of the training range but that subset range is not seen during training).
> --> There is zero overlap between the test-set, OOD Easy and OOD Hard test sets. We had incorrectly explained the tests sets in Table 1 of the Appendix. We have now corrected the Table also added clarifications on Figure 9. We are sorry for the confusion. Overall, we have constructed our dataset so that the training and OOD test-sets are disjoint (Figures 2 & 9 help by visualizing this). Effectively, this guarantees that for all sequences, at least one parameter is outside of the training range. We have also added clarification for the Video Pendulum dataset in the Section A.2 of the appendix, which along with Figure 2, should give a clear picture of the OOD nature of the dataset. It is also important to note that for example in Figure 9 the area of the blue region is about half of the green region (again there is no overlap). This means that the parameter space in the OOD test sets is far from constrained or trivially different. Furthermore, in dynamical systems a small variation in parameters can produce widely different trajectories in the phase space (e.g. bifurcations). Under that light the differences between the datasets are quite substantial.
>
> Thank you for the suggestion regarding interpolation, this is a regime were models can usually cope well so we excluded it.
>
> * Have the authors tried Gaussian distribution (correspondingly L2 loss) for the decoder? I wonder how the results might differ from the Laplace distribution.
> --> We tried this and L1 worked better for all models.
>
> * The prediction quality is reported by using perceptual metrics LPIPS and SSIM. These metrics compare the deep feature space and statistical properties of the images respectively. I think these metrics are not sufficient for evaluating the predictions of the dynamic. If it is possible then kindly report RMSE and/or NLL.
> --> We also include PSNR in the appendix. These 3 metrics (LPIPS, SSIM and PSNR) are some of the most commonly used metrics in the literature in video dynamics but we consider adding RMSE/NLL in future versions.
>
> * Fig1 caption: Do the input-output dimensions differ? I don't think the labeling in the figure is correct as there are some inconsistencies. For e.g. in a single time step:
> --> Yes input/output dimensions can be different, this is frames as a prediction and not a reconstruction task.
>
> Thank you very much for spotting the typos, they have been corrected in the updated manuscript.
>
> We hope that we have addressed at least some of the reviewers concerns, especially we want to reiterate that the datasets are OOD without overlapping. We also hope to have provided some context for the results that will make assessing their significance easier.

---

### Author Response · Authors · 2021-11-18
**Authors note on two important issues: OOD setting and results significance**

We would like to address two issues that have been raised by the reviewers.

1) The first issue is whether our setting is indeed OOD.
We would like to stress that there is zero overlap between the test-set, OOD Easy and OOD Hard test sets. We had incorrectly explained the tests sets in Table 1 of the Appendix. We have now corrected the Table also added clarifications on Figure 9. We are sorry for the confusion. Overall, we have constructed our dataset so that the training and OOD test-sets are disjoint (Figures 2 & 9 help by visualizing this). Effectively, this guarantees that for all sequences, at least one parameter is outside of the training range. We have also added clarification for the Video Pendulum dataset in the Section A.2 of the appendix, which along with Figure 2, should give a clear picture of the OOD nature of the dataset. It is also important to note that for example in Figure 9 the area of the blue region is about half of the green region (again there is no overlap). This means that the parameter space in the OOD test sets is far from constrained or trivially different. Furthermore, in dynamical systems a small variation in parameters can produce widely different trajectories in the phase space (e.g. bifurcations). Under that light the differences between the datasets are quite substantial.

2) Secondly we would like to give some context that will hopefully make assessing their significance easier.
In Table 8 in the Appendix we present the numerical values of the RMSE. For the OOD datasets, VAE-SD consistently reduces the error in comparison to the VAE by 5-22%.
For the RSSM (Table 10 in Appendix) there is a 8-16% LPIPS reduction across datasets, 5-9% PSNR increase and approx 1-3% increase in SSIM.
We would like to give some context why we believe these results are quite important:
1. the improvement is against a SoTA dynamical prediction model (RSSM). Improving on it up to 16% in some metrics should not be considered trivial. For example in Saxena et al. (Table 1; NeurIPS 2021), the improvement over RSSM is less than 5% (when there is any).
2. they are consistent across datasets
3. they come over the whole range of predicted frames (800 timesteps in video pendulum)
4. come from quite diverse in and out-of-distribution data-sets.

---

### Decision · Program_Chairs · 2022-01-20

**Decision:**

Reject

**Comment:**

This manuscript tackles an interesting and significant line of research of long-term prediction and out-of-distribution generalization in time series models. I strongly believe this problem is an important one to solve. However, in its current form, its novelty is marginal, and the experiments fail to decisively show advantages. It also lacks of systematic improvements and error analysis. Further work could make it ready for publication at a next conference.